# Toward Robust Multilingual Adaptation of LLMs for Low-Resource Languages

Haolin Li [* 1 2 3]   Haipeng Zhang [* 2]   Mang Li [2]   Yaohua Wang [1]   Lijie Wen [3]   Yu Zhang [2]   Biqing Huang [1]

## Abstract

Large language models (LLMs) continue to struggle with low-resource languages due to limited training data, translation noise, and unstable cross-lingual alignment. We propose LiRA (Linguistic Robust Anchoring for LLMs), a plug-and-play framework that improves multilingual adaptation through fine-tuning on existing pretrained backbones. LiRA combines two components: Arca, which aligns low-resource inputs to a shared English semantic space through anchor-based alignment and collaborative encoding, and LaSR, a language-aware head that promotes cross-lingual consistency for retrieval, ranking, and reasoning. We provide a theoretical analysis showing that, under bounded anchoring error and translation-induced bias, LiRA yields bounded representation deviation and stable downstream behavior under local Lipschitz continuity. We further introduce a multilingual product retrieval dataset covering five Southeast Asian and two South Asian languages. Experiments on retrieval, ranking, question answering, and reasoning benchmarks show that LiRA consistently improves strong multilingual and LLM-based baselines. Code and data will be released publicly.

## 1. Introduction

Large language models (LLMs) have made remarkable progress in natural language understanding and reasoning. Yet these gains are unevenly distributed: performance is concentrated in high-resource languages such as English and Chinese, while low-resource languages (LRLs) continue to lag far behind. This gap is driven by long-tailed pretraining

distributions (Haddow et al., 2022) (Figure 1a), limited or noisy parallel data (Ataman et al., 2025), and unstable cross-lingual alignment (Xu et al., 2025). As a result, directly deploying LLMs for retrieval and reasoning in LRLs often leads to degraded accuracy and brittle, inconsistent behavior, limiting the promise of truly inclusive NLP systems.

Existing cross-lingual adaptation methods typically fall into two families: machine translation (MT)-based pipelines (Artetxe et al., 2023; Shubham, 2024) and multilingual approaches (Singh et al., 2024). Although MT-based pipelines can be effective, they are prone to error propagation (Wu et al., 2019; Shen et al., 2023) and semantic drift (Beinborn & Choenni, 2020; Ataman et al., 2025), especially for complex settings that require multi-step reasoning or nuanced interpretation. Multilingual encoders, in contrast, offer more language-agnostic representations but often fail to inherit the strong English-centric reasoning ability exhibited by LLMs (Hu et al., 2020; Yoon et al., 2024). Recent systems such as MindMerger (Huang et al., 2024) attempt to narrow this gap by integrating the capabilities of LLMs and multilingual models, yet its reliance on parallel translation data may still lead to error propagation and semantic drift. Similarly, LUSIFER (Man et al., 2025) connects a multilingual encoder with an LLM-based embedding model to enable cross-lingual transfer; however, its training paradigm lacks information from low-resource languages, which may compromise the performance of the transfer during inference. Moreover, despite empirical gains, these lines of work remain theoretically underdeveloped and do not consistently close the gap on key low-resource tasks such as retrieval, ranking, and reasoning.

Our work is motivated by the observation that cross-lingual adaptation still relies heavily on machine translation pipelines or multilingual encoders. In real-world settings, such as low-resource e-commerce, these solutions are often undermined by translation noise and unstable cross-lingual representations. More importantly, they rarely model how semantic drift and representation-mapping errors propagate through the system, making robustness difficult to analyze and even harder to improve systematically. To address these challenges, we introduce **LiRA** (**Li**nguistic **R**obust **A**nchoring), a *plug-and-play* framework that anchors low-resource languages to an English semantic space while preserving LLM-level reasoning (Figure 1b). LiRA can be

[1]Department of Automation, Tsinghua University, Beijing, China [2]Alibaba International Digital Commerce Group, Beijing, China [3]School of Software, Tsinghua University, Beijing, China. Correspondence to: Lijie Wen <wenlj@tsinghua.edu.cn>, Yu Zhang <zhangyu@alibaba-inc.com>, Biqing Huang <hbq@tsinghua.edu.cn>.

*Proceedings of the 43rd International Conference on Machine Learning*, Seoul, South Korea. PMLR 306, 2026. Copyright 2026 by the author(s).

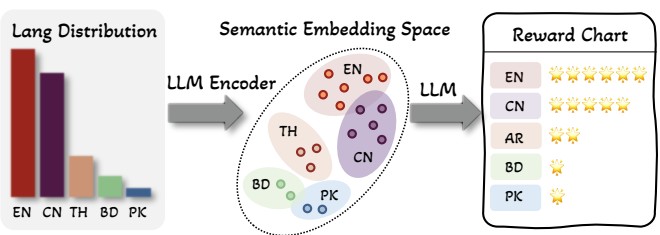
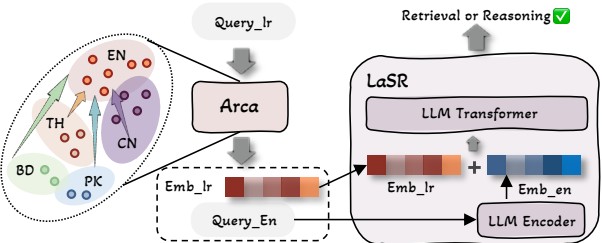

(a) Long-tailed pre-training distribution leads to large cross-language performance disparity.

(b) Overview of our model framework. lr and en represent low-resource language and English, respectively.

*Figure 1.* Challenge and overview of our LiRA.

attached to a variety of pretrained backbones and requires only lightweight fine-tuning to improve cross-lingual representations. Unlike prior approaches, LiRA is grounded in a rigorous theoretical framework that provides formal guarantees on the completeness and stability of the learned representations. LiRA integrates two complementary components: (i) *Arca*, which reduces semantic drift by aligning multilingual representations with English via critic–actor interaction and feature anchoring; and (ii) *LaSR*, a lightweight head that fuses multilingual and English embeddings with queue-based objectives to support robust retrieval and reasoning in low-resource settings. In this way, LiRA leverages strong English capabilities and transfers them effectively to underrepresented languages.

Our main contributions are summarized as follows:

- We propose **LiRA**, a *plug-and-play* cross-lingual framework that transfers LLMs' strong English capability to mid- and low-resource languages.
- We establish solid theoretical foundations, providing rigorous guarantees of LiRA's completeness and stability.
- We release a product retrieval dataset covering 5 Southeast Asian and 2 South Asian mid- and low-resource languages, enabling further research in this underexplored area.
- Extensive experiments across ranking, retrieval, and reasoning tasks demonstrate that LiRA achieves new state-of-the-art performance.

## 2. Related Work

**Cross-lingual Information Retrieval** Early CLIR systems relied on multilingual PLMs (e.g., mBERT (Devlin et al., 2019), XLM-R (Conneau et al., 2020)) for alignment; recent work moves toward supervision-light transfer and LLM-embedding adaptation. LUSIFER integrates a multilingual encoder with an English-specialized LLM-based embedding model via a connector to yield strong zero-shot multilingual retrieval (Man et al., 2025). However, the zero-shot training paradigm limits further performance gains. Beyond

passage-level retrieval, CCPR formulates contextualized *phrase*-level cross-lingual retrieval to mitigate polysemy (Li et al., 2024), while XRAG benchmarks end-to-end cross-lingual RAG from retrieval to generation (Liu et al., 2025). Domain resources (e.g., CrossMath) and synthetic datasets (e.g., SWIM-IR) further broaden evaluation and training coverage (Gore et al., 2024; Thakur et al., 2024). Existing methods often rely on heuristic designs without a unified theoretical framework, which may restrict their generality and theoretical interpretability. By contrast, our approach is grounded in a principled framework that unifies multilingual latent embedding spaces, mitigates semantic drift, and supports plug-and-play modular training. Furthermore, cross-lingual contamination can inflate reported performance (Yao et al., 2024), prompting us to propose new real-world data for fair comparison.

**LLM-based Cross-lingual Reasoning** Recent studies (Cueva et al., 2024) have intensified attention to reasoning tasks in low-resource language settings (Kim et al., 2023). MindMerger merges external multilingual understanding into LLMs and trains collaborative use of internal reasoning and external language skills, substantially boosting reasoning in non-English settings (Huang et al., 2024). However, MindMerger relies heavily on parallel translation corpora, which may introduce semantic drift due to translation noise. Recent multilingual reasoning methods further improve non-English reasoning through instruction tuning, preference optimization, and staged adaptation. For example, mCoT enhances reasoning consistency via multilingual instruction tuning, MAPO formulates multilingual alignment as preference optimization, and LinguaLIFT adopts a two-stage instruction-tuning framework for low-resource language reasoning (Lai & Nissim, 2024; She et al., 2024; Zhang et al., 2026). Process-level improvements—such as Chain-of-Preference Optimization and Chain-of-Code—provide transferable reasoning scaffolds that can be combined with language anchoring strategies (Zhang et al., 2024; Li et al., 2023). Mechanism-focused studies reveal cross-lingual knowledge barriers and even "cross-lingual collapse" of reasoning traces toward dominant pretraining languages;

mitigation includes mixed-language finetuning, reward shaping for language consistency, and representation steering to strengthen non-English token representations (Chua et al., 2024; Zhao et al., 2024; Park et al., 2025; Mahmoud et al., 2025). Recent surveys systematize multilingual reasoning evaluations and resources (Ghosh et al., 2025). These methods typically rely on a single embedding-space alignment primarily designed for reasoning tasks, and their performance may not transfer well to broader settings such as ranking and retrieval. In contrast, our proposed Arca minimizes translation-induced semantic drift, provides better interpretability, and can be extended to support multiple tasks—including reasoning, ranking, and retrieval—thereby improving overall generalization.

## 3. Theoretical Foundations

### 3.1. Preliminaries

**Setup.** We study cross-lingual retrieval/reasoning under low-resource inputs. Let $\mathcal{X}$ be the low-resource language (LRL) sentence space and $\mathcal{Y}$ the English sentence space. For each $x \in \mathcal{X}$, an LLM-based translator $T : \mathcal{X} \to \mathcal{Y}$ produces an observed translation $y = T(x)$, and we introduce an (unobserved) *ideal* translation $y^\star \in \mathcal{Y}$ that is semantically equivalent to $x$. We consider two representation paths into a shared $d$-dimensional space: a trained multilingual encoder $g : \mathcal{X} \to \mathbb{R}^d$ that embeds $x$ directly into an English-aligned semantic space, and an English encoder $h : \mathcal{Y} \to \mathbb{R}^d$ that embeds English sentences. Our system forms the concatenated representation $\mathbf{z}(x) := [\, g(x);\ h(y)\,] \in \mathbb{R}^{2d}$, which is then fed into a downstream $f$ (e.g., for retrieval, ranking or reasoning). For analysis, we define the ideal reference representation $\mathbf{z}^\star(x) := [\, h(y^\star);\ h(y^\star)\,] \in \mathbb{R}^{2d}$, corresponding to the target behavior where both paths agree on the same English semantics; our goal is to bound the representation deviation $\|\mathbf{z}(x) - \mathbf{z}^\star(x)\|_2$ and the induced deviation $\|f(\mathbf{z}(x)) - f(\mathbf{z}^\star(x))\|_2$. See A.3 for the theoretical explanation of why we concatenate the two representation paths.

**Assumption 1** (Semantic Anchoring)**.** For all $x \in \mathcal{X}$, the mismatch between the anchor and the English encoding of its translation is bounded:

$$\|g(x) - h(y^*)\|_2 \le \epsilon_1, \qquad \epsilon_1 \ge 0.$$

**Assumption 2** (Translation fidelity)**.** Let $s$ be a latent semantic variable with conditionals $p(s \mid x)$ and $p(s \mid y)$. The translator $T$ preserves semantics up to $\epsilon_2$ in KL:

$$D_{\mathrm{KL}}\big(p(s \mid x) \,\|\, p(s \mid T(x))\big) \le \epsilon_2, \qquad \epsilon_2 \ge 0.$$

**Definition 1** (RKHS representation)**.** We model $h(y)$ as the kernel mean embedding (KME) of the semantic distribution $p(s \mid y)$ into an RKHS $\mathcal{H}$ induced by a positive–definite kernel $k$:

$$h(y) = \mu_{p(s|y)} := \mathbb{E}_{s \sim p(s|y)}\big[\varphi(s)\big], \quad \varphi(s) := k(s, \cdot).$$

The kernel embedding maps any probability measure $p$ over the semantic space into the RKHS specified by $k$ (i.e., $\mu_p = \mathbb{E}_{s \sim p}[\varphi(s)]$). For bounded inputs, the kernel satisfies

$$0 < k(s, s) = \big\langle k(s, \cdot),\, k(s, \cdot)\big\rangle_{\mathcal{H}} \le C^2,$$

for some constant $C > 0$. See Appendix A.2 for details.

**Definition 2** (Data-local Lipschitzness)**.** On a finite discrete domain, any encoder admits a (local) Lipschitz constant. Concretely, over the dataset $\mathcal{Y}_{\mathrm{data}}$, we define the data-local Lipschitz constant at $y$ (with neighborhood radius $\delta$) as

$$L^{\mathrm{loc}}(y; \delta) := \max_{y' \in \mathcal{N}_\delta(y)} \frac{\big\|f_{\mathrm{LLM}}(\mathbf{z}) - f_{\mathrm{LLM}}(\mathbf{z}^\star)\big\|_2}{\big\|\mathbf{z} - \mathbf{z}^\star\big\|_2}.$$

Here $\mathbf{z} = [\, g(x);\ h(y)\,]$, $\mathcal{N}_\delta(y) = \{\, y' \in \mathcal{Y}_{\mathrm{data}} : 0 < d_{\mathrm{tok}}(y, y') \le \delta \,\}$ denotes the token-edit neighborhood (we use $\delta = 1$ in most experiments). We also report the empirical $q$-quantile $L^{(q)}(\delta)$, where $q \in (0, 1)$ is the quantile level; $L^{(q)}(\delta)$ is the $q$-th empirical quantile of $L^{\mathrm{loc}}(y; \delta)$ over $y \in \mathcal{Y}_{\mathrm{data}}$. ; for example, with $q = 0.95$ we observe $L^{(0.95)} \approx 0.034$. See Appendix A.2 for details.

### 3.2. Theorem

**Theorem** (Representation deviation) Under Assumptions 1–2 and Definitions 1–2, let $\mathbf{z}^\star = [\, h(y^\star);\ h(y^\star)\,]$ and $\mathbf{z} = [\, g(x);\ h(y)\,]$. Then

$$\|\mathbf{z} - \mathbf{z}^\star\|_2 \le \epsilon_1 + C\sqrt{2\,\epsilon_2}. \tag{1}$$

**Corollary** (Downstream stability) For $f_{\mathrm{LLM}}$ that is **locally Lipschitz** with constant $L^{\mathrm{loc}}(y; \delta)$ as in Definition 2, we have

$$\big\|f_{\mathrm{LLM}}(\mathbf{z}) - f_{\mathrm{LLM}}(\mathbf{z}^\star)\big\|_2 \le L^{\mathrm{loc}}(y; \delta)\,\big(\epsilon_1 + C\sqrt{2\,\epsilon_2}\big). \tag{2}$$

As $\epsilon_1, \epsilon_2 \to 0$, we obtain

$$\|\mathbf{z} - \mathbf{z}^\star\|_2 \to 0 \quad \text{and} \quad \|f_{\mathrm{LLM}}(\mathbf{z}) - f_{\mathrm{LLM}}(\mathbf{z}^\star)\|_2 \to 0.$$

The theorem and its corollary imply that, by minimizing $\epsilon_1$ and $\epsilon_2$, the model obtains high-fidelity and robust representations that effectively support downstream tasks. Full proofs of the theorem and corollary are provided in Appendix A.1.

### 3.3. Theory assumptions and practical validity.

Our theoretical analysis does not require "ideal translations" or "ideal semantic anchoring" to hold in practice. Instead, it only models two ubiquitous sources of error in cross-lingual alignment: (i) representation mapping error between an ideal target vector $z^*$ and the observed representation $z$, and (ii) translation-induced noise. Here, $z^*$ is introduced purely as a mathematical reference for $z$—not as a strict

real-world prerequisite—and the assumptions are used to derive objectives that explicitly target semantic drift and vector-level misalignment. Empirically, we observe that the gap between $z$ and $z^*$ (measured by similarity) decreases as training proceeds, and the loss consistently drops across different Arca training tasks, supporting the practical relevance of our theoretical formulation. See D.2 for empirical measurements during the training process.

# 4. Method

## 4.1. Arca

Let $\mathcal{X}$ be a low–resource source space and $\mathcal{Y}$ the English space. For any $x \in \mathcal{X}$, a translator $T : \mathcal{X} \to \mathcal{Y}$ yields $y = T(x)$. We introduce two representation paths: an *anchoring map* $g : \mathcal{X} \to \mathbb{R}^d$ that lands source sentences directly in an "English semantic" space, and an English encoder $h : \mathcal{Y} \to \mathbb{R}^d$. We concatenate $\mathbf{z} = [g(x); h(y)]$ and score it with an LLM critic. Arca aims to reduce the two terms appearing in our generalization bound (Sec. 3.2): the *anchoring error* $\epsilon_1$ and the *translation distortion* $\epsilon_2$. Arca (Figure 2) comprises three modules: (i) a *Translation Critic* that judges candidates with semantic/emotional/pragmatic scores; (ii) an *Embedding Critic* that anchors feature paths to translation paths via a regression-style penalty; and (iii) an *Actor* trained with policy gradients that fuses both critics.

### 4.1.1. FEATURE ANCHORING (MINIMIZING $\epsilon_1$)

Given $x \in \mathcal{X}$ with observed translation $y = T(x) \in \mathcal{Y}$, we align the multilingual path $g(x)$ and the English path $h(y)$ by resolving tokenizer-length and dimensional mismatches via temporal pooling and a shared Adaptor. Let $g_{\text{tok}}(x) \in \mathbb{R}^{L_x \times d_g}$ and $h_{\text{tok}}(y) \in \mathbb{R}^{L_y \times d_h}$ denote the token-level streams before sentence pooling. We apply a temporal pooling operator with $S_{\text{feat}}$ bins (a hyperparameter) to obtain fixed-length sequences:

$$
\begin{aligned}
G(x) &= P_{S_{\text{feat}}}\big(g_{\text{tok}}(x)\big) \in \mathbb{R}^{S_{\text{feat}} \times d_g}, \\
H(y) &= P_{S_{\text{feat}}}\big(h_{\text{tok}}(y)\big) \in \mathbb{R}^{S_{\text{feat}} \times d_h}.
\end{aligned} \tag{3}
$$

A single shared Adaptor $A(\cdot)$ maps either side into a common $d$-dimensional space (with internal projections when $d_g \neq d_h$). We then pool along the temporal axis to form sentence-level vectors:

$$
\begin{aligned}
E_{lr} &= \text{pool}\big(A(G(x))\big) \in \mathbb{R}^d, \\
E_{en} &= \text{pool}\big(A(H(y))\big) \in \mathbb{R}^d.
\end{aligned} \tag{4}
$$

The feature-anchoring objective contracts the path discrepancy by cosine alignment:

$$
\mathcal{L}_{\text{anchor}} = 1 - \cos\big(E_{lr}, E_{en}\big). \tag{5}
$$

Minimizing Eq. (5) reduces the anchoring radius $\epsilon_1$ after length normalization (Eq. (3)) and feature alignment (Eq. (4)).

### 4.1.2. TRANSLATION CRITIC (MINIMIZING $\epsilon_2$)

Given a source $x$ and its candidate set $\{y_k\}_{k=1}^K$, a lightweight LLM judge produces three calibrated scores $s_k, e_k, p_k \in [1, 10]$ for *semantic fidelity*, *emotional consistency*, and *pragmatic tone*. We collect

$$
\mathbf{r}_k = [\, s_k, \ e_k, \ p_k \,]^\top.
$$

*Role for $\epsilon_2$.* These scores probe adequacy and well-formedness of $y_k$ with respect to $x$, serving as a proxy for small semantic divergence between $p(s \,|\, x)$ and $p(s \,|\, y_k)$; maximizing their contribution in the policy drives smaller $\epsilon_2$.

### 4.1.3. OVERALL OBJECTIVE

For each candidate we form the policy feature by concatenating the critic scores with the adaptor similarity:

$$
\mathbf{c}_k = [\, s_k, \ e_k, \ p_k, \ \text{sim}_k \,]^\top. \tag{6}
$$

A small MLP produces logits $g_\phi(\mathbf{c}_k)$ and the policy $\pi_\phi(k \,|\, \mathbf{c}_{1:K}) = \text{softmax}([g_\phi(\mathbf{c}_1), \dots, g_\phi(\mathbf{c}_K)])_k$. We use the composite reward

$$
R_k = 0.1 \cdot (\alpha s_k + \beta e_k + \gamma p_k) + \delta \, \text{sim}_k, \tag{7}
$$

sample $a \sim \pi_\phi$, and optimize with REINFORCE:

$$
\mathcal{L}_{\text{RL}} = -\mathbb{E}_{a \sim \pi_\phi}[R_a] \approx -\log \pi_\phi(a \,|\, \mathbf{c}_{1:K}) \cdot R_a. \tag{8}
$$

The full Arca objective is

$$
\mathcal{L} = \mathcal{L}_{\text{RL}} + \eta \, \mathcal{L}_{\text{anchor}}. \tag{9}
$$

Here, $\mathcal{L}_{\text{anchor}}$ reduces the anchoring error $\epsilon_1$, while $\mathcal{L}_{\text{RL}}$ favors low-distortion candidates, shrinking $\epsilon_2$—jointly tightening the bound in Sec. 3.2.

## 4.2. LaSR

Given any-language text, a multilingual encoder yields $E_{\text{lr}}$ while a shared English encoder (prompted LLM encoder) yields $E_{\text{en}}$. $E_{\text{lr}}$ denotes the text embedding in the low-resource language, and $E_{\text{en}}$ denotes the embedding of the corresponding English text. We fuse $E_{\text{en}}$ and $E_{\text{lr}}$ into a single $\ell_2$-normalized embedding used for all ranking, retrieval and reasoning tasks. Training is supported by two FIFO buffers: (i) *CorrQueue* for correlation-based objectives under small batches, and (ii) *DocQueue* for listwise nDCG with in-language negatives. Both queues are stop-grad for cached entries and are updated in FIFO manner with a maximum size $K$.

**Encoders.** Each branch follows "tokenize → encoder → pooling": $E_{\text{en}}$ and $E_{\text{lr}}$, as shown in Eq. (4). Pooling is fixed

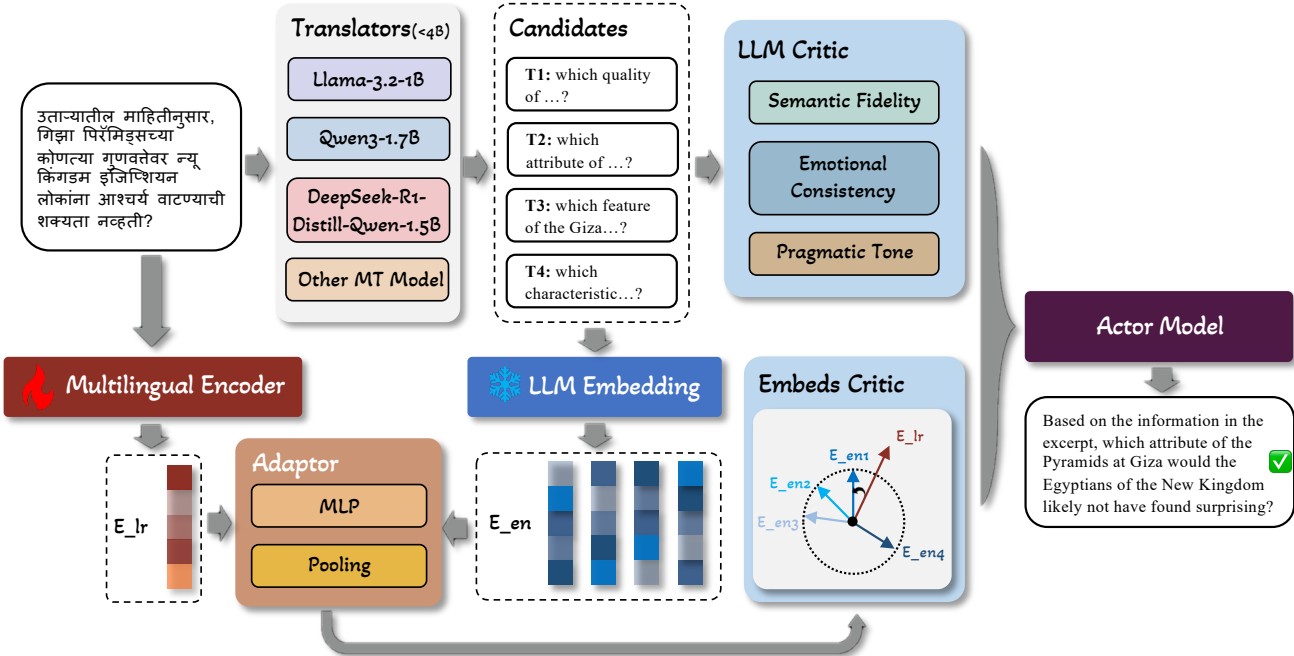

*Figure 2.* An overview of Arca.

and the two streams are concatenated and linearly projected before entering the LLM Transformer. The transformer attends across the two streams and returns a fused hidden vector $\mathbf{z}$, which is then normalized $\hat{\mathbf{z}} = \mathbf{z}/\|\mathbf{z}\|$.

**Training steps (Figure 3b).** Each optimization step processes two mini-batches: (1) query batch and (2) document batch. We forward them through the shared encoders and the LLM Transformer to obtain $\{\hat{\mathbf{z}}_q\}$ and $\{\hat{\mathbf{z}}_d\}$ and compute $s(q, d)$ for the task at hand. Two FIFO buffers are updated in the background:

- **CorrQueue (Rank task).** Caches tuples (Pred, Gold) produced on ranking datasets (e.g., STS). At step $t$ we concatenate the current predictions/labels with up to $K$ cached pairs (detached, no gradient) and compute a *correlation loss*: $\mathcal{L}_{\text{corrQ}} = \alpha (1 - \text{Pearson}) + (1 - \alpha)(1 - \text{Soft-Spearman})$. See Appendix D.5 for details.
- **DocQueue (Retrieval task).** Caches (doc_id, language, $\hat{\mathbf{z}}_d$) from recent steps for *in-language* hard-negative mining. Given a query, we form a candidate list (1 positive + mined negatives), compute differentiable ranks and a *soft nDCG@k* objective $\mathcal{L}_{\text{ndcg}} = 1 - \text{nDCG@}k$. To avoid mislabeled near negatives, we apply a temperatured down-weighting on very similar non-positives ("safe negatives"), and add two light regularizers (top-1 hinge, mean/variance control) to stabilize training: $\mathcal{L}_{\text{retr}} = \mathcal{L}_{\text{ndcg}} + \lambda_h \mathcal{L}_{\text{hinge}} + \lambda_r \mathcal{L}_{\text{mv}}$. See Appendix D.5 for details.

### 4.3. Motivation and Discussion.

A key challenge in low-resource multilingual reasoning is the *representation imbalance* across languages: modern LLMs typically acquire substantially stronger reasoning capability in high-resource languages (notably English), where reasoning-oriented supervision is abundant, while comparable signals are scarce in many low-resource languages. Under this data constraint, our goal is to make the most of the strong reasoning capabilities that LLMs have acquired in high-resource languages and *transfer* them to underrepresented languages in a robust and analyzable manner.

LiRA addresses the imbalance by explicitly modeling and shrinking two ubiquitous sources of cross-lingual shift that degrade downstream retrieval and reasoning: (i) **semantic drift** caused by translation noise and linguistic variation, and (ii) **representation mapping error** between multilingual and English embedding spaces. Arca reduces both shifts via anchor-based alignment and critic–actor training, which aims to *mitigate* translation-induced noise rather than propagate it. Importantly, LiRA is *plug-and-play*: it can be attached to different pretrained backbones with lightweight fine-tuning, and it does not assume ideal translations in practice. On top of Arca, LaSR performs a two-way coupling between multilingual and English representations with queue-based objectives, balancing cross-lingual alignment and task-driven discrimination so that English-centric reasoning signals can be leveraged without destabilizing multilingual embeddings.

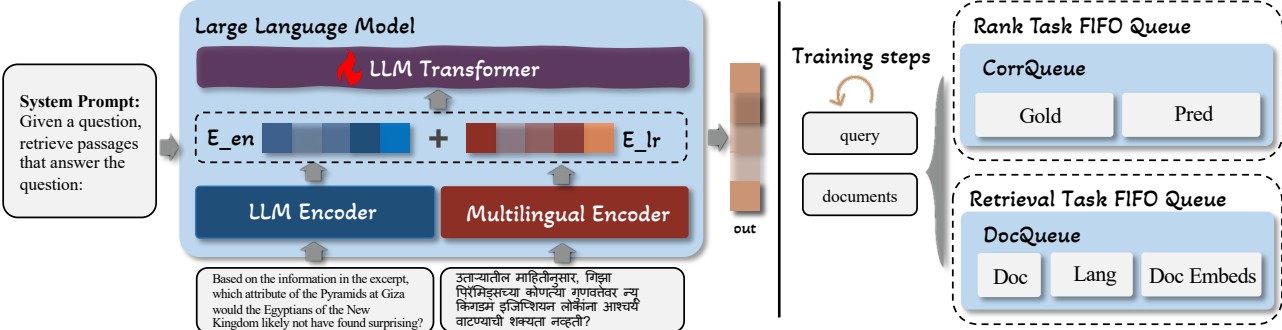

(a) An overview of LaSR—given any-language input, shared encoders produce English and multilingual features that are fused by a lightweight transformer into a single normalized embedding (q/d) for retrieval and ranking.

(b) Two FIFO buffers: CorrQueue caches (pred, gold) for correlation-based objectives; DocQueue caches (doc id, lang, embed) for listwise nDCG with in-language negatives.)

*Figure 3.* An overview of LaSR.

*Table 1.* Evaluation on our new LazRetrieval dataset. **bold** indicates the best result; underlined indicates the second best.

| METHOD | | PARAMS | BN | ID | MS | UR | TH | PH | VI | AVG. |
|---|---|---|---|---|---|---|---|---|---|---|
| SENTENCE-T5-XXL | *2021* | 4.8B | 34.11 | 71.77 | 49.19 | 27.84 | 23.58 | 84.20 | 28.61 | 44.56 |
| GTR-XXL | *2021* | 4.8B | 34.85 | 75.92 | 49.36 | 30.39 | 22.94 | 84.94 | 46.15 | 48.17 |
| SIMCSE | *2021* | 330M | 31.76 | 66.71 | 43.27 | 27.68 | 32.32 | 74.00 | 44.16 | 44.88 |
| CONTRIEVER | *2022* | 110M | 39.95 | 74.95 | 48.90 | 35.71 | 15.43 | 83.74 | 64.75 | 51.00 |
| GTE-LARGE | *2023* | 335M | 39.36 | 77.59 | 51.88 | 36.76 | 17.21 | 87.65 | 65.34 | 52.61 |
| BGE-EN-V1.5 | *2023* | 335M | 41.06 | 78.78 | 53.28 | 37.52 | 18.35 | 88.18 | 68.72 | 54.09 |
| E5-LARGE-V2 | *2024* | 335M | 41.04 | 78.78 | 53.77 | 37.22 | 17.63 | 87.24 | 61.31 | 52.84 |
| E5-MISTRAL-7B | *2024* | 7.24B | 48.27 | 75.43 | 71.01 | 53.62 | 61.75 | 83.18 | 65.44 | 64.51 |
| QWEN3-E-0.6B | *2025* | 0.6B | 38.36 | 63.95 | 62.37 | 40.73 | 55.28 | 74.38 | 59.46 | 56.31 |
| **WITH LIRA-LARGE** | **OURS** | 8.5B | 48.60 | 74.43 | 71.26 | 49.84 | 66.39 | 83.90 | 70.67 | 66.44 |
| QWEN3-E-4B | *2025* | 4B | 49.81 | 68.92 | 71.45 | 49.30 | 73.39 | 78.47 | 68.31 | 65.66 |
| **WITH LIRA-LARGE** | **OURS** | 11.9B | 56.24 | 75.29 | 77.03 | 55.69 | 77.72 | 84.81 | 74.92 | 71.67 |
| QWEN3-E-8B | *2025* | 8B | 50.59 | 76.01 | 73.36 | 50.37 | 69.67 | 84.78 | 71.56 | 68.05 |
| **WITH LIRA-BASE** | **OURS** | 14.4B | 52.91 | 76.59 | 75.20 | 52.30 | 71.11 | 85.92 | 73.23 | 69.61 |
| **WITH LIRA-LARGE** | **OURS** | 15.9B | 57.70 | 77.33 | 77.93 | 56.67 | 78.52 | 86.03 | 75.86 | 72.86 |
| **WITH LIRA-MAX** | **OURS** | 103B | **66.30** | **78.53** | **81.54** | **68.53** | **83.12** | **87.44** | **78.48** | **77.71** |

## 5. Experiment

### 5.1. Experimental Details

All experiments are conducted on a single server with $4\times$ A100-80GB GPUs. We evaluate LiRA on three families of tasks to assess generality: (i) retrieval, measured by nDCG@10; (ii) sentence ranking, measured by Pearson correlation; and (iii) reading comprehension and mathematical reasoning, both measured by accuracy. For retrieval and sentence ranking we use Qwen3-Embedding-8B as the backbone encoder. For reading comprehension and mathematical reasoning we use Qwen3-8B as the backbone model. Following Sec. 3.2, we prefer backbones with a smaller Lipschitz constant $L$ to tighten error propagation; we thus select a comparatively more stable LLM as the backbone (see Appendix A.2 and Table 7). Model choices, full hyperparameters and additional implementation details are reported in Appendix C and D, including Arca translation selection strategy (C.1), prompt engineering (C.2) and bad case anal-

ysis (C.3). As discussed in D.3, we analyze training and inference efficiency in detail. When translations are precomputed offline, LiRA incurs little additional wall-clock overhead during training or inference.

Due to potential discrepancies in pretraining data, we ensured fairness by fine-tuning all models used in our experiments on each dataset with identical training hyperparameters before evaluation. Interestingly, we observed that fine-tuning Qwen3 brought no gains on existing public datasets. This phenomenon may be attributed to Qwen3 having already seen portions of these public datasets during its pretraining phase. For the critic, the weights are set to $\alpha, \beta, \gamma, \delta = 0.4, 0.4, 0.3, 1.0$.

We instantiate LiRA at three parameter scales. LiRA-Base uses three lightweight MT models—OPUS-MT, m2m100-418M, and nllb-200-600M—together with Qwen3-1.7B, which also serves as the critic. LiRA-Large upgrades the translator set to Qwen3-1.7B, DeepSeek-R1-Distill-

*Table 2.* Evaluation on main public retrieval datasets, and the effect of adding LiRA-Max to representative embedding backbones. **bold** indicates the best; underlined indicates the second best. Column abbreviations: MLQA-R = MLQARetrieval, Bele-R = BelebeleRetrieval.

| METHOD | | MLQA-R | BELE-R | STS22 | AVG. |
|---|---|---|---|---|---|
| SIMCSE | *2021* | 7.41 | 18.35 | 37.95 | 21.24 |
| ST5-XXL | *2021* | 20.82 | 41.68 | 59.02 | 40.51 |
| GTR-XXL | *2021* | 20.19 | 38.02 | 60.11 | 39.44 |
| CONTRIEVER | *2022* | 9.75 | 22.94 | 41.72 | 24.80 |
| GTE-LARGE | *2023* | 16.99 | 31.82 | 53.79 | 34.20 |
| ↪ WITH LIRA | | **21.43** | **38.29** | **59.17** | **39.63** |
| BGE-EN-1.5 | *2023* | 16.64 | 31.19 | 50.77 | 32.87 |
| ↪ WITH LIRA | | **21.01** | **35.64** | **53.94** | **36.83** |
| E5-LARGE | *2024* | 17.04 | 31.12 | 54.31 | 34.16 |
| E5-MISTRAL | *2024* | 31.54 | 54.75 | 71.37 | 52.55 |
| ↪ WITH LIRA | | **34.23** | **57.24** | **76.55** | **56.01** |
| LUSIFER | *2025* | 36.68 | 57.81 | 70.49 | 54.99 |
| QWEN3-E-8B | *2025* | 81.13 | 85.94 | 71.64 | 79.57 |
| ↪ **WITH LIRA** | | **82.01** | **87.03** | **75.00** | **81.35** |

Qwen-1.5B, and Llama3.2-1B-Instruct, with Qwen3-1.7B again acting as the critic model. Finally, LiRA-Max employs three large LLMs—Qwen3-32B, DeepSeek-R1-Distill-Qwen-32B, and Gemma-2-27B—where Qwen3-32B is used as the critic. See D.1 for more detailed information.

Our analysis predicts that two noise can be amplified by the backbone's local sensitivity (captured by a local Lipschitz-type factor). We estimate $L_{\mathrm{emp}}(\delta)$ for Qwen3-Embedding backbones (Table 7) and observe that larger models are consistently more stable. This stability ranking matches the retrieval trend on LazRetrieval (A.2): performance increases from Qwen3-E-0.6B→4B→8B, and LiRA built on more stable backbones yields larger gains (LiRA-Large on 4B/8B > on 0.6B; LiRA-Max further improves the average). Overall, Table 7–A.2 provide empirical evidence that reducing local sensitivity mitigates error amplification and improves downstream retrieval.

We note that the larger parameter count of LiRA mainly comes from auxiliary translators and critics used for offline candidate generation and selection. These components are not fully trainable adaptation modules, and translations can be pre-computed and reused. Therefore, the term lightweight" refers to trainable adaptation and post-preprocessing inference rather than the total number of auxiliary parameters involved during preprocessing.

### 5.2. Datasets

We use standard public datasets: *BelebeleRetrieval* (Bandarkar et al., 2024), *MLQARetrieval* (Enevoldsen et al., 2025), *STS22* (Enevoldsen et al., 2025), *MGSM* (Shi et al., 2022), and *X-CSQA* (Lin et al., 2021). The first two are retrieval benchmarks, *STS22* evaluates sentence-level corre-

lation, and *MGSM/X-CSQA* assess mathematical reasoning and reading comprehension, respectively. As we have observed above that contamination in cross-lingual training data may inflate the reported performance, we additionally introduce a new real-world dataset to enable a fair comparison. We release a de-identified e-commerce retrieval dataset, LazRetrieval, and a larger companion set, LazRetrieval-mega for further research. Both cover seven languages from South and Southeast Asia: Vietnamese (Vi), Thai (Th), Indonesian (Id), Malay (Ms), Urdu (Ur), Bengali (Bn), and Filipino/Tagalog (Ph). LazRetrieval contains **10 k** examples per language; LazRetrieval-mega contains **1,000 k** examples per language and is intended for pretraining/supporting large-scale adaptation. Unless otherwise noted, our experiments use *LazRetrieval*. Since *MGSM* and *X-CSQA* have no training split in our setup, we evaluate in a zero-shot fashion: the Arca is trained on *BelebeleRetrieval*, while the lightweight LaSR head is left untrained for these tasks. Detailed information about the datasets can be found in B.

### 5.3. Results Analysis

**Retrieval & sentence ranking.** On public benchmarks (Table 2), LiRA consistently improves the base model (Qwen3-E-8B) on all three metrics: MLQARetrieval 82.01 vs. 81.13 (+0.88), BelebeleRetrieval 87.03 vs. 85.94 (+1.09), and STS22 75.00 vs. 71.64 (+3.36), yielding a higher macro average 81.35 (+1.78). On our new LazRetrieval-70K (Table 1), Qwen3-Embedding with LiRA-Large also improves the average from 68.05 to 72.86 (+4.81). The gains are particularly pronounced on relatively low-resource locales (e.g., Bn +7.11, Ur +6.30, Th +8.85), suggesting that the combination of English-space anchoring for $g(x)$ and the LaSR head helps attenuate translation and representation noise. We also observe better rank-correlation, matching our design of queue-augmented CorrQ and listwise soft-nDCG.

**Scaling behaviour on LazRetrieval.** We observe a consistent scaling trend across Qwen3 embedding backbones of different sizes (Table 1). For the smallest backbone (Qwen3-E-0.6B), attaching LiRA-Large yields substantial gains on all languages, indicating that LiRA can effectively compensate for limited model capacity. A similar pattern holds for Qwen3-E-4B, where LiRA-Large improves Bn from 49.81 to 56.24 and Ur from 49.30 to 55.69. For the full 8B backbone, LiRA forms a smooth performance–capacity curve: the average score increases from 68.05 to 69.61 with LiRA-Base, 72.86 with LiRA-Large, and 77.71 with LiRA-Max, accompanied by consistent improvements on all seven LazRetrieval languages. Notably, the relative gains are larger for smaller encoders, which is consistent with the hypothesis that LiRA mitigates cross-lingual noise rather than simply adding capacity. Overall, these results demonstrate that LiRA provides monotonic benefits across parameter scales and is particularly effective for smaller backbones.

*Table 3.* MGSM accuracy (%). **bold** indicates the best; underlined indicates the second best.

| METHOD | PARAMS | YEAR | BN | TH | SW | JA | ZH | DE | FR | RU | ES | EN | AVG. |
|---|---|---|---|---|---|---|---|---|---|---|---|---|---|
| MONOREASON | 7B | 2024 | 6.8 | 7.2 | 6.8 | 36.4 | 38.4 | 55.2 | 54.4 | 52.0 | 57.2 | 68.8 | 38.3 |
| MULTIREASON-LORA | 7B | 2022 | 29.6 | 35.2 | 28.0 | 52.0 | 54.8 | 59.6 | 58.4 | 62.4 | 59.6 | 64.8 | 50.4 |
| MULTIREASON-SFT | 7B | 2024 | 33.2 | 40.0 | 42.0 | 42.0 | 42.0 | 45.2 | 44.8 | 45.2 | 48.0 | 52.0 | 43.4 |
| QALIGN | 7B | 2024 | 39.6 | 40.4 | 44.0 | 44.0 | 48.4 | 54.8 | 56.8 | 52.4 | 59.6 | 68.0 | 49.6 |
| LANGBRIDGE | 7B | 2024 | 42.8 | 50.4 | 43.2 | 40.0 | 45.2 | 56.4 | 50.8 | 52.4 | 58.0 | 63.2 | 50.2 |
| TRANSLATE-EN | 3.3B | 2023 | 48.4 | 37.6 | 37.6 | 49.2 | 46.8 | 60.4 | 56.4 | 47.6 | 59.6 | 65.5 | 50.6 |
| mCoT | 7B | 2024 | 65.6 | 67.6 | 67.2 | 65.2 | 64.8 | 61.2 | 63.8 | 66.8 | 68.4 | 71.6 | 66.2 |
| MAPO | 13B | 2024 | 44.8 | 47.6 | 55.2 | 56.0 | 59.6 | 59.2 | 62.8 | 59.2 | 63.6 | 71.6 | 58.0 |
| LINGUALIFT | 7B | 2026 | 63.0 | 66.6 | 64.2 | 52.0 | 58.0 | 69.6 | 62.8 | 71.6 | 75.2 | 76.0 | 65.5 |
| MINDMERGER-HARD | 10.7B | 2024 | 46.0 | 36.0 | 48.4 | 52.4 | 54.4 | 60.4 | 56.0 | 60.4 | 62.0 | 71.2 | 54.7 |
| MINDMERGER-SOFT | 10.7B | 2024 | 50.4 | 52.8 | 57.2 | 54.4 | 53.6 | 61.2 | 57.6 | 60.8 | 58.4 | 66.8 | 57.3 |
| QWEN3-8B | 8B | 2025 | 66.4 | 69.3 | 59.1 | 64.5 | 67.9 | **71.0** | 69.3 | 72.5 | **75.1** | 78.9 | 69.4 |
| **WITH LiRA-LARGE** | 15.9B | OURS | **69.6** | **72.5** | **61.9** | **69.1** | **70.3** | 69.3 | **69.8** | **73.9** | 75.0 | **79.2** | **71.1** |

*Table 4.* X-CSQA accuracy (%). **bold** indicates the best; underlined indicates the second best.

| METHOD | | SW | UR | HI | AR | VI | JA | PL | ZH | NL | RU | IT | DE | PT | FR | ES | EN | AVG. |
|---|---|---|---|---|---|---|---|---|---|---|---|---|---|---|---|---|---|---|
| TRANSLATE-EN | 2023 | 36.5 | 41.3 | 48.4 | 44.6 | 51.8 | 47.1 | 53.3 | 51.5 | 55.0 | 56.3 | 57.3 | 54.7 | 57.2 | 55.5 | 71.3 | 71.3 | 52.3 |
| MULTIREASON-LORA | 2022 | 25.1 | 32.0 | 39.2 | 42.2 | 56.6 | 55.9 | 60.6 | 62.2 | 61.3 | 62.8 | 66.3 | 64.9 | 66.2 | 67.4 | 67.7 | 79.3 | 56.9 |
| MULTIREASON-SFT | 2024 | 27.6 | 29.2 | 32.0 | 28.7 | 38.8 | 38.7 | 45.5 | 43.8 | 45.9 | 46.5 | 50.2 | 49.1 | 51.2 | 52.1 | 54.3 | 67.2 | 43.8 |
| MONOREASON | 2024 | 24.2 | 25.1 | 32.9 | 32.3 | 50.9 | 49.1 | 50.6 | 56.5 | 57.5 | 56.0 | 56.0 | 61.2 | 61.7 | 63.5 | 64.0 | 76.3 | 51.3 |
| QALIGN | 2024 | 35.1 | 32.6 | 37.8 | 36.3 | 50.5 | 49.2 | 57.1 | 54.8 | 56.3 | 58.3 | 58.3 | 58.8 | 59.8 | 60.3 | 63.1 | 75.7 | 52.3 |
| LANGBRIDGE | 2024 | 31.8 | 30.5 | 30.6 | 30.6 | 33.3 | 33.9 | 39.8 | 39.8 | 38.4 | 39.1 | 37.4 | 36.4 | 33.8 | 38.2 | 38.8 | 44.4 | 36.1 |
| MINDMERGER-HARD | 2024 | 33.1 | 29.9 | 40.4 | 37.7 | 52.9 | 49.9 | 54.7 | 55.4 | 58.0 | 59.7 | 58.6 | 61.9 | 62.5 | 63.6 | 75.2 | 75.2 | 53.1 |
| MINDMERGER-SOFT | 2024 | 45.5 | 46.2 | 48.4 | 51.4 | 60.6 | 53.9 | 63.3 | 62.9 | 63.8 | **66.8** | 67.0 | 67.1 | 68.1 | 69.1 | **75.2** | 78.1 | 61.0 |
| LINGUALIFT | 2026 | **46.7** | 46.2 | 50.6 | 54.1 | 60.4 | 53.8 | 62.9 | 64.1 | 63.5 | 63.5 | 65.7 | 68.7 | 68.2 | 68.7 | 67.9 | 78.3 | 61.5 |
| QWEN3-8B | 2025 | 35.7 | 51.6 | 52.8 | 60.9 | 63.0 | 59.3 | 62.5 | 66.6 | 64.7 | 64.1 | 67.6 | 66.9 | 68.2 | 69.8 | 70.1 | 82.8 | 62.9 |
| **WITH LiRA-LARGE** | OURS | 40.8 | **52.7** | **55.6** | **63.9** | **65.0** | **61.3** | **64.2** | **68.3** | **67.9** | 66.3 | **69.7** | **70.8** | **72.0** | **70.7** | 74.6 | **84.2** | **65.5** |

**Mathematics.** On MGSM (Table 3), LiRA brings a small but consistent gain over Qwen3-8B, with an average score of 69.4 vs. 71.1 (+1.7). Per-language analysis shows improvements or matches on 8/10 languages (e.g., Bn/Th/Zh), while performance on several high-resource languages remains comparable (De, Es). This indicates that anchoring low-resource languages to an English semantic space effectively improves reasoning robustness without compromising performance on high-resource languages.

**Comprehension.** On X-CSQA (Table 4), LiRA outperforms Qwen3-8B on 15/16 languages and raises the average from 62.9 to 65.5 (+2.6). Improvements concentrate on lower-resource or typologically distant languages (Ur/Hi/Vi), consistent with our motivation to leverage English capabilities and transfer them to underrepresented languages, thereby improving multilingual reasoning.

**Cross-backbone robustness.** Table 2 also evaluates LiRA as a pluggable module on three representative encoders. Across MLQA Retrieval, BelebeleRetrieval, and STS22, LiRA consistently improves over the corresponding backbones. The averaged gains over the three tasks are positive for all backbones tested, suggesting that the effect is not tied to a single encoder family.

## 5.4. Ablation

The ablation results in Table 5 demonstrate the contribution of each component in LiRA. Removing the LLM Critic or Embeds Critic leads to the most significant performance drop, particularly on Pearson correlation and accuracy, highlighting the importance of dual-level critics for effective supervision. The translation and multilingual encoder modules also provide consistent gains, showing their role in enhancing cross-lingual generalization. Finally, eliminating the FIFO loss queue results in the largest degradation on nDCG@10, confirming its necessity in stabilizing optimization. Overall, each component is essential, and their synergy ensures the robustness of LiRA across tasks. In our ablation study, the three reported metrics are obtained on different tasks: nDCG@10 is evaluated on LazRetrieval, Pearson is evaluated on STS22, and Acc. is evaluated on MGSM.

We further examine whether the gains come merely from adding auxiliary pretrained models. As shown in Appendix D.4, a naive dual-path combination without LiRA training does not improve over the backbone, suggesting that explicit anchoring and consistency training are necessary.

## 5.5. Supplementary Experiments

To offer additional insight into the practical behavior of our framework, we provide supplementary analyses in the appendix. In particular, we report (i) an empirical break-

*Table 5.* Ablation study of LiRA on retrieval, sentence ranking, and reasoning tasks.

| METHOD | NDCG@10 | PEARSON | ACC. |
|---|---|---|---|
| LiRA (FULL) | **77.71** | **75.00** | **71.1** |
| ↪ – LLM CRITIC | 71.29 | 72.19 | 68.9 |
| ↪ – EMBEDS CRITIC | 65.77 | 61.78 | 67.3 |
| ↪ – TRANSLATIONS | 75.48 | 74.39 | 70.5 |
| ↪ – MULTI-ENC | 75.59 | 72.43 | 69.5 |
| ↪ – FIFO QUEUE | 64.29 | 69.82 | – |

down of ARCA's translation candidate selections across datasets (Appendix C, Figure 10), which helps reveal potential evaluator-induced preferences; (ii) estimates of the RKHS boundedness surrogate $\widehat{C}$ across backbone scales (Appendix A, Table 6); and (iii) data-local Lipschitz statistics $L_{\mathrm{emp}}(\delta)$ under token-level perturbations (Appendix A, Table 7). Together, these results provide practical guidance for instantiating our theoretical bounds and interpreting model behavior beyond the primary benchmarks.

# 6. Conclusion

We proposed LiRA, a framework for robust multilingual LLM adaptation that unifies retrieval, sentence ranking, and reasoning tasks under a common anchoring principle. By combining anchored representations with critic-guided alignment and queue-based objectives, LiRA consistently improves over strong Qwen3 baselines across both public benchmarks and our newly introduced LazRetrieval dataset. Ablation studies further validate the complementary contributions of each component. We hope our dataset and framework can inspire future work on multilingual LLM adaptation.

# Impact Statement

This work aims to improve multilingual access to large language model capabilities for low-resource languages. It may benefit users and communities whose languages are underrepresented in current NLP systems, particularly in retrieval and reasoning applications. Potential risks include propagation of translation biases, uneven performance across languages, and misuse in multilingual commercial settings. We encourage careful evaluation across languages, domains, and user groups before deployment.

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

# A. Theoretical Details

## A.1. Math Proof

**Theorem 1** (Representation deviation bound). *Under Assumptions 1–2 and Definitions 1–2, let the optimal English representation be* $\mathbf{z}^\star = [\,h(y^\star);\ h(y^\star)\,]$ *and the framework output be* $\mathbf{z} = [\,g(x);\ h(y)\,]$. *Then*

$$\big\|\mathbf{z} - \mathbf{z}^\star\big\|_2 \ \leq\ \epsilon_1 \ +\ C\,\sqrt{2\,\epsilon_2}, \tag{10}$$

*where* $C > 0$ *is the kernel boundedness constant from Definition 1, i.e.,* $\sup_s k(s,s) \leq C^2$. *This constant reflects the geometry of the semantic RKHS; smaller* $C$ *indicates more stable embeddings. In practice,* $C$ *can be estimated empirically on a corpus (e.g.,* $C \approx 0.6867$ *in our experiments).*

*Proof.* Let $y^\star$ be an *ideal translation* such that $p(s \mid x) = p(s \mid y^\star)$. By block structure and the triangle inequality,

$$\big\|\mathbf{z} - \mathbf{z}^\star\big\|_2 = \left\|\begin{bmatrix} g(x) - h(y^\star) \\ h(y) - h(y^\star) \end{bmatrix}\right\|_2 \ \leq\ \|g(x) - h(y^\star)\|_2 + \|h(y) - h(y^\star)\|_2. \tag{11}$$

By Assumption 1,

$$\|g(x) - h(y^\star)\|_2 \ \leq\ \|g(x) - h(y)\|_2 + \|h(y) - h(y^\star)\|_2 \ \leq\ \epsilon_1 + \|h(y) - h(y^\star)\|_2. \tag{12}$$

Next, by Assumption 2 and Pinsker's inequality,

$$\|p(s \mid y) - p(s \mid y^\star)\|_1 \ \leq\ \sqrt{2\,D_{\mathrm{KL}}\big(p(s \mid y) \,\|\, p(s \mid y^\star)\big)} \ \leq\ \sqrt{2\,\epsilon_2}. \tag{13}$$

Using the RKHS mean-embedding view of $h$ (Definition 1) and the bounded-kernel assumption (see, e.g., (Sriperumbudur et al., 2010)),

$$\begin{aligned}
\|h(y) - h(y^\star)\|_2 &= \big\|\mu_{p(s|y)} - \mu_{p(s|y^\star)}\big\|_{\mathcal{H}} \\
&\leq\ C\,\|\,p(\cdot \mid y) - p(\cdot \mid y^\star)\,\|_{\mathrm{TV}} \\
&=\ \frac{C}{2}\,\|p(s \mid y) - p(s \mid y^\star)\|_1 \\
&\leq\ C\,\sqrt{\tfrac{1}{2}\,D_{\mathrm{KL}}\big(p(s \mid y) \,\|\, p(s \mid y^\star)\big)} \ \leq\ C\,\sqrt{\tfrac{\epsilon_2}{2}}.
\end{aligned} \tag{14}$$

Plugging (14) into (12) and then into (11) yields

$$\big\|\mathbf{z} - \mathbf{z}^\star\big\|_2 \ \leq\ \left(\epsilon_1 + C\sqrt{\tfrac{\epsilon_2}{2}}\right) + C\sqrt{\tfrac{\epsilon_2}{2}} \ =\ \epsilon_1 + C\sqrt{2\,\epsilon_2},$$

which proves the claim. $\square$

**Corollary 1.** *Downstream stability.* Let $f_{\mathrm{LLM}}$ *denote the downstream scorer. If* $f_{\mathrm{LLM}}$ *is locally Lipschitz around* $[g(x); h(y)]$ *with constant* $L^{\mathrm{loc}}(y; \delta)$ *as in Definition 2, then*

$$\big\|f_{\mathrm{LLM}}(\mathbf{z}) - f_{\mathrm{LLM}}(\mathbf{z}^\star)\big\|_2 \ \leq\ L^{\mathrm{loc}}(y; \delta)\left(\epsilon_1 + C\,\sqrt{2\,\epsilon_2}\right). \tag{15}$$

*Proof.* By the (local) Lipschitz property of $f_{\mathrm{LLM}}$ and Theorem 3.2,

$$\big\|f_{\mathrm{LLM}}(\mathbf{z}) - f_{\mathrm{LLM}}(\mathbf{z}^\star)\big\|_2 \ \leq\ L^{\mathrm{loc}}(y; \delta)\,\|\mathbf{z} - \mathbf{z}^\star\|_2 \ \leq\ L^{\mathrm{loc}}(y; \delta)\left(\epsilon_1 + C\sqrt{2\,\epsilon_2}\right).$$

$\square$

**Instantiation.** In our measurements we obtain $L^{(0.95)}(y; \delta) \approx 0.034$ and $C \approx 0.6867$ (representation dimension $n = 4096$). A representative bound (reported in $\ell_1$ for readability) is

$$\big\|f_{\mathrm{LLM}}(\mathbf{z}) - f_{\mathrm{LLM}}(\mathbf{z}^\star)\big\|_1 \ \leq\ 0.034 \cdot \left(\epsilon_1 + 1.9423\,\sqrt{\epsilon_2}\right). \tag{16}$$

## A.2. About Definition

**Definition 1**  (RKHS representation) Let $h : \mathcal{Y} \rightarrow \mathbb{R}^d$ denote the English sentence encoder. We view $h(y)$ as the *kernel mean embedding* (KME) of the conditional semantic distribution $p(s \mid y)$ in an RKHS $(\mathcal{H}, k)$:

$$h(y) \;=\; \mu_{p(s|y)} \;=\; \mathbb{E}_{s \sim p(s|y)}\big[\varphi(s)\big], \qquad \varphi(s) = k(s, \cdot). \tag{17}$$

The kernel $k$ is assumed bounded on the (semantics) domain: $0 < k(s,s) = \langle k(s,\cdot), k(s,\cdot) \rangle_{\mathcal{H}} \leq C^2$ for some constant $C > 0$.

*Remark* 1 (On estimating the boundedness constant $C$). For any probability measure $P$ on the input space with $x, x' \overset{\text{i.i.d.}}{\sim} P$,

$$\big\| \mu_P \big\|_{\mathcal{H}}^2 = \mathbb{E}_{x,x'}\big[k(x,x')\big] \;\leq\; \mathbb{E}_x\big[k(x,x)\big], \tag{18}$$

where the inequality follows from $k(x,x') \leq \sqrt{k(x,x)\,k(x',x')}$ for PSD kernels and Jensen. Thus $\|h(y)\|_{\mathcal{H}} = \|\mu_{p(s|y)}\|_{\mathcal{H}}$ provides a *lower-bound proxy* for $\mathbb{E}[k(x,x)]$, but it does not identify the *pointwise* upper bound $\sup_s k(s,s) = C^2$. In practice one may report empirical surrogates (e.g., corpus-wise maxima of $\|h(y)\|$), while the theoretical $C$ remains a kernel-dependent constant. See Smola et al. (2007); Shioda et al. (2017); Yoshikawa et al. (2015) for background.

**Estimator.**  Let $E \in \mathbb{R}^{V \times d}$ be the model's input embedding table and $y = (w_1, \ldots, w_T)$ the tokenized sentence with attention mask $m_t \in \{0,1\}$. We compute the *unnormalized* mean-pooled sentence vector

$$\widehat{h}(y) \;=\; \frac{1}{\sum_{t=1}^{T} m_t} \sum_{t=1}^{T} m_t\, E[w_t, :] \;\in\; \mathbb{R}^d, \qquad \text{and its norm } \|\widehat{h}(y)\|_2. \tag{19}$$

The corpus-level estimators are

$$\widehat{C}_{\max} \;=\; \max_{y \in \mathcal{Y}_{\text{probe}}} \|\widehat{h}(y)\|_2, \qquad \widehat{C}_q \;=\; \text{Quantile}_{y \in \mathcal{Y}_{\text{probe}}}\big(\|\widehat{h}(y)\|_2, \; q\big), \tag{20}$$

where $q \in (0,1)$ (e.g., $q = 0.90, 0.95, 0.99$) provides robust surrogates. By construction $\widehat{C}_{\max} \leq C$ (a *lower* bound on the true $C$).

**Implementation.**  We follow the released script `compute_C_rkhs.py`: (i) tokenize each sentence, (ii) fetch token embeddings via `get_input_embeddings()`, (iii) mean-pool with the attention mask (no $\ell_2$ normalization), (iv) take $\| \cdot \|_2$ and aggregate statistics (*max*, mean, std, median, $p90$, $p95$, $p99$, sample count). Unless otherwise noted, we probe on STS22 (`sentence1`) with `max_length=8192` and report per-model results in Table 6.

*Table 6.* Estimates of the RKHS bound $C$ on STS22 (field: *sentence1*) using mean-pooled *unnormalized* input embeddings (no $\ell_2$ post-normalization). $\widehat{C}_{\max} = \max_y \|\widehat{h}(y)\|_2$; $\widehat{C}_q$ denotes the $q$-quantile.

| Model | $\widehat{C}_{\max}$ | Mean | Std | Median | P90 | P95 | P99 |
|---|---|---|---|---|---|---|---|
| Qwen3-Embedding-0.6B | 0.5333 | 0.1944 | 0.0221 | 0.1878 | 0.2240 | 0.2376 | 0.2605 |
| Qwen3-Embedding-4B | 0.5457 | 0.2073 | 0.0324 | 0.1971 | 0.2465 | 0.2672 | 0.3302 |
| Qwen3-Embedding-8B | 0.6866 | 0.3988 | 0.0434 | 0.3900 | 0.4529 | 0.4752 | 0.5503 |

**Analysis.**  Table 6 and Figure 4 shows that the empirical RKHS bound surrogates $\widehat{C}$ increase moderately with model size (0.6B→4B→8B), which tightens separation in representation space but enlarges the worst-case radius $r(C) = \epsilon_1 + 2\,C\sqrt{2\epsilon_2}$ when plugging $C$ into our bounds. Because $\widehat{C}_{\max} \leq C$ (Definition A.2), any bound instantiated with $\widehat{C}_{\max}$ or $\widehat{C}_q$ is *optimistic* (it may understate the true worst case); therefore we recommend using (i) a *high-probability* bound using $C = \widehat{C}_{0.95}$ together with its empirical coverage on validation, and (ii) a *worst-case* bound using $C = \widehat{C}_{\max}$ as a lower envelope for the true $C$. For rigor, one can calibrate a multiplicative slack $\kappa \geq 1$ by back-testing—choose the smallest $\kappa$ such that the inequality with $C = \kappa\,\widehat{C}_{0.95}$ holds on at least 95% of held-out samples. Finally, $\widehat{C}$ is sensitive to tokenization length and domain; we thus compute $\widehat{C}$ on the target corpus (STS22 *sentence1* by default) with unnormalized mean pooling, and advise re-estimating it in-domain when the deployment distribution shifts.

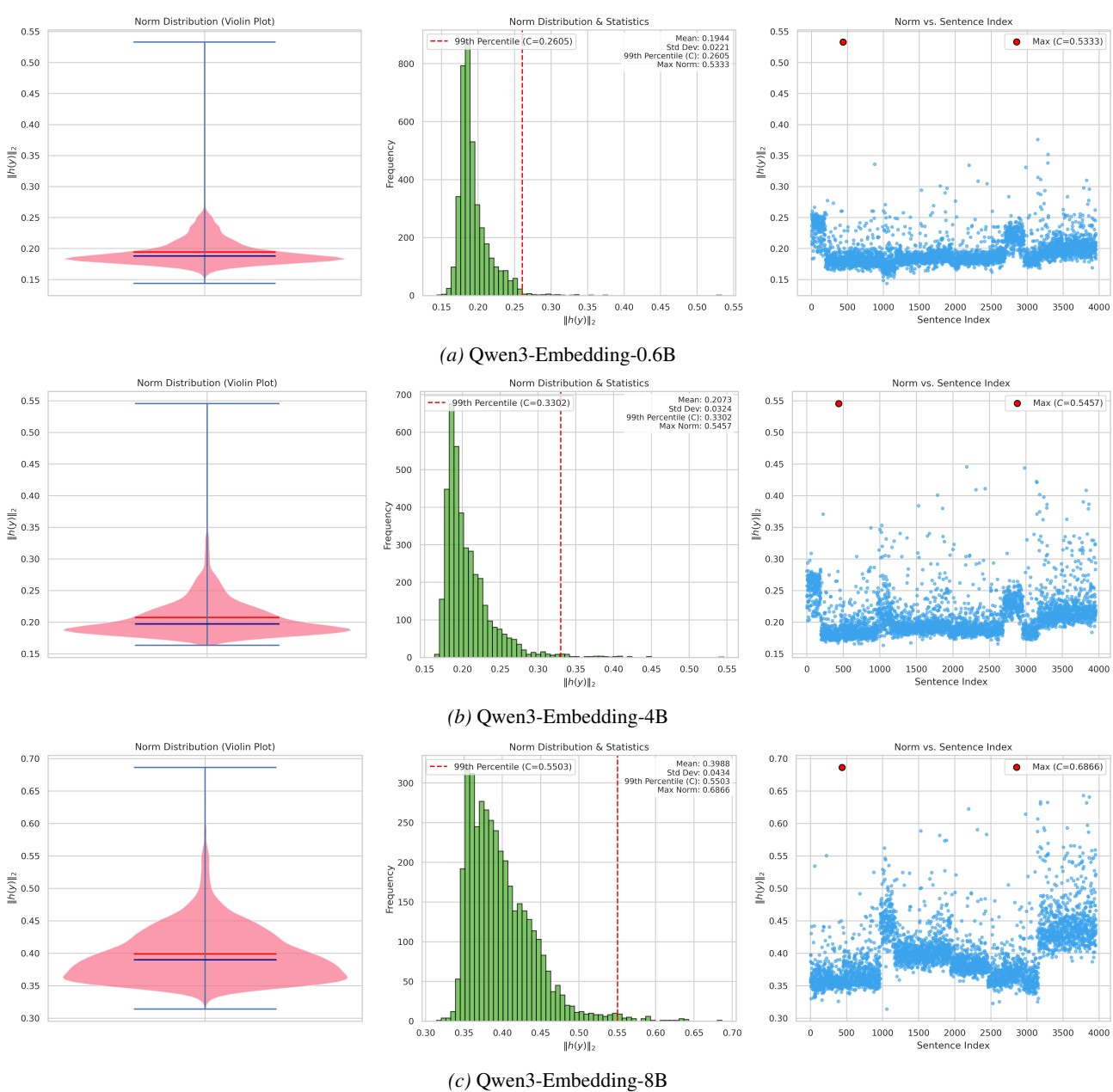

*(a)* Qwen3-Embedding-0.6B

*(b)* Qwen3-Embedding-4B

*(c)* Qwen3-Embedding-8B

*Figure 4.* Empirical estimates of the RKHS bound $C$ across model scales. Each subfigure shows (left) a violin plot of $\|\widehat{h}(y)\|_2$, (middle) a histogram with the 99th percentile marked, and (right) a scatter of norms by sentence index.

**Definition 2** (Data-local Lipschitz constant) "How fast can the encoder's output change under small edit perturbations of a sentence in real text data?" By standard Lipschitz-continuity arguments on finite discrete domains, any encoder admits a Lipschitz constant. Hence, on the dataset $\mathcal{Y}_{\text{data}}$ the encoder satisfies

$$L_h^{\text{loc}}(y; \delta) = \max_{y' \in \mathcal{N}_\delta(y)} \frac{\left\| f_{\text{LLM}}(\mathbf{z}) - f_{\text{LLM}}(\mathbf{z}^\star) \right\|_2}{\left\| \mathbf{z} - \mathbf{z}^\star \right\|_2}, \tag{21}$$

where $\mathbf{z} = [\, g(x);\, h(y) \,]$. We denote its $q$-quantile by $L_h^{(q)}$, measured by the script described earlier (e.g., $q = 0.95$, $L_h^{(0.95)} \approx 0.05$). *Note.* $\mathcal{N}_\delta(y)$ is the neighborhood defined by token-level edit distance $\leq \delta$; in practice we use $\delta = 1$.

*Table 7.* Empirical local Lipschitz estimates $L_{\text{emp}}(\delta)$ (percent). We report mean, std, median, and high quantiles (P90/P95/P99), plus max.

| Model | $\delta$ | Mean | Std | Median | P90 | P95 | P99 | Max |
|---|---|---|---|---|---|---|---|---|
| | 1 | 6.8% | 6.2% | 5% | 13.7% | 18.1% | 31.5% | 58.6% |
| | 2 | 6.5% | 5.4% | 5.1% | 12.7% | 16.3% | 27.2% | 55.7% |
| Qwen3-Embedding-0.6B | 3 | 5.7% | 4.8% | 4.4% | 11.3% | 14.6% | 23.9% | 44.9% |
| | 5 | 4.2% | 3.8% | 3.1% | 8.5% | 11.1% | 18.8% | 34% |
| | 8 | 2.7% | 3.3% | 1.7% | 5.8% | 8% | 16% | 77% |
| | 10 | 2.2% | 2.9% | 1.3% | 4.9% | 6.9% | 14.6% | 55.7% |
| | 1 | 5.5% | 5% | 4.1% | 11.1% | 14.5% | 27% | 54.9% |
| | 2 | 5.2% | 4% | 4.2% | 10% | 12.5% | 21% | 41.4% |
| Qwen3-Embedding-4B | 3 | 4.6% | 3.7% | 3.7% | 8.8% | 11.3% | 18% | 38.2% |
| | 5 | 3.4% | 3% | 2.6% | 7% | 8.7% | 14.5% | 38.7% |
| | 8 | 2.2% | 2.6% | 1.5% | 4.8% | 6.7% | 12.4% | 29.4% |
| | 10 | 1.8% | 2.4% | 1.1% | 4.1% | 5.7% | 11.9% | 39.2% |
| | 1 | 3.4% | 3% | 2.5% | 6.4% | 8.6% | 15.8% | 30.3% |
| | 2 | 3.2% | 2.6% | 2.5% | 6.1% | 7.9% | 14.3% | 29.3% |
| Qwen3-Embedding-8B | 3 | 2.9% | 2.3% | 2.2% | 5.5% | 7.1% | 12.2% | 22.7% |
| | 5 | 2.1% | 1.9% | 1.6% | 4% | 5.3% | 9.5% | 21.7% |
| | 8 | 1.4% | 1.7% | 0.9% | 2.9% | 4.1% | 8.3% | 20.4% |
| | 10 | 1.1% | 1.6% | 0.7% | 2.4% | 3.5% | 7.9% | 30% |

**Explanation.** The *data-local Lipschitz constant* is defined w.r.t. the downstream scorer $f_{\text{LLM}}$ as

$$L_h^{\text{loc}}(y; \delta) = \max_{y' \in \mathcal{N}_\delta(y)} \frac{\left\| f_{\text{LLM}}(\mathbf{z}) - f_{\text{LLM}}(\mathbf{z}^\star) \right\|_2}{\left\| \mathbf{z} - \mathbf{z}^\star \right\|_2}, \tag{22}$$

is defined as follows.

- $f_{LLM}$: is any fixed downstream model (e.g., Qwen-3, Qwen-3-Embedding, BERT).

- $d_{\text{tok}}(y, y')$ is the token-level edit distance between two sentences (e.g., Levenshtein distance).

**1) $\delta$-neighborhood (in the corpus).** For a finite corpus $\mathcal{Y}_{\text{data}}$ and any sentence $y$, define the radius-$\delta$ neighborhood

$$\mathcal{N}_\delta(y) = \left\{\, y' \in \mathcal{Y}_{\text{data}} \;:\; 0 < d_{\text{tok}}(y, y') \leq \delta \,\right\}.$$

That is, $\mathcal{N}_\delta(y)$ contains all sentences that differ from $y$ by at most $\delta$ token edits (e.g., by exactly one token when $\delta = 1$).

**2) Pointwise local Lipschitz constant.** The quantity $L_h^{\text{loc}}(y; \delta)$ measures the encoder's rate of change within the data neighborhood. As long as $y$ has at least one neighbor, this value is finite.

**3) Empirical quantile over the dataset.** For a confidence level $q \in (0, 1)$ (e.g., $q = 0.95$), define

$$L_h^{(q)}(\delta) = \text{Quantile}_{y \in \mathcal{Y}_{\text{data}}} \left( L_h^{\text{loc}}(y; \delta),\, q \right). \tag{23}$$

In words, $q \cdot 100\%$ of sentences in the corpus have local Lipschitz constants no greater than $L_h^{(q)}$. Empirically, for $\delta$ between 1 and 10, the local Lipschitz ratios of Qwen3-Embedding are below 0.1 for 99% of samples; moreover, as $\delta$ increases the ratios decrease and concentrate, indicating that the local Lipschitz constant both exists and is measurable.

**Experimental notes.**

- **Lipschitz Ratio** ($L_h$). For an original sentence $s$ and its perturbed version $s'$, the ratio is

$$L_h = \frac{\left\|\text{Embed}(s) - \text{Embed}(s')\right\|_2}{\text{EditDistance}(s, s')}.$$

  Here $\|\cdot\|_2$ is the Euclidean norm between embedding vectors, and $\text{EditDistance}$ is the Levenshtein distance (minimum number of single-character edits to transform $s$ into $s'$). A small and stable ratio indicates local stability/robustness: small input changes do not cause large embedding shifts. If the ratio grows markedly with $\delta$, the model may vary sharply in some regions.

- $\delta$ **(Delta).** The maximum number of edit operations allowed for the perturbation. The script evaluates multiple $\delta$ values (e.g., $1, 2, 3, 5$, etc.).

- **Mean.** The average of the ratios at a fixed $\delta$, reflecting the typical sensitivity at that perturbation level.

- **Standard Deviation.** The dispersion of the ratios; larger values indicate greater variability across sentences/perturbations.

- **Quantiles.** E.g., median (50%), 90%, 95%, and 99% quantiles. The 95% quantile means that $95\%$ of ratios are no greater than that value, useful for detecting rare but high-sensitivity cases.

- **Sample Count.** The number of valid ratios computed for a given $\delta$ (cases with no change after perturbation are excluded).

By tracking these statistics as $\delta$ varies, we assess the encoder's local Lipschitz characteristics and, in turn, its stability and robustness.

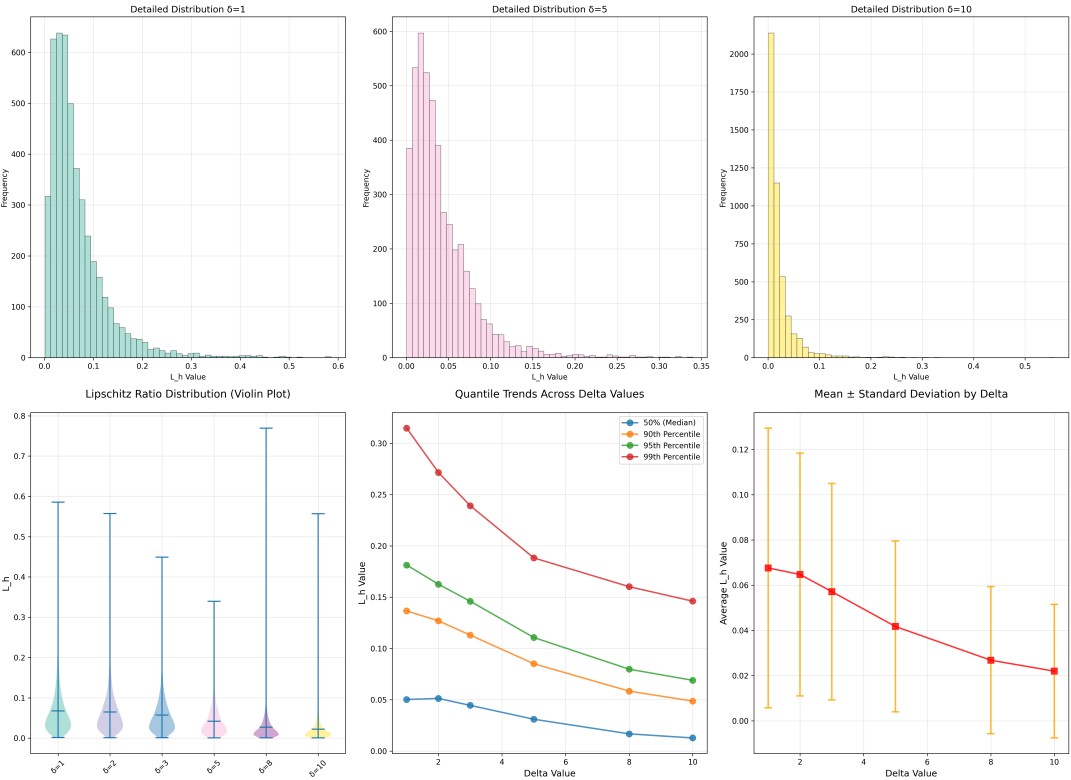

*Figure 5.* Qwen3-Embedding-0.6B: Local Lipschitz analysis across $\delta$.

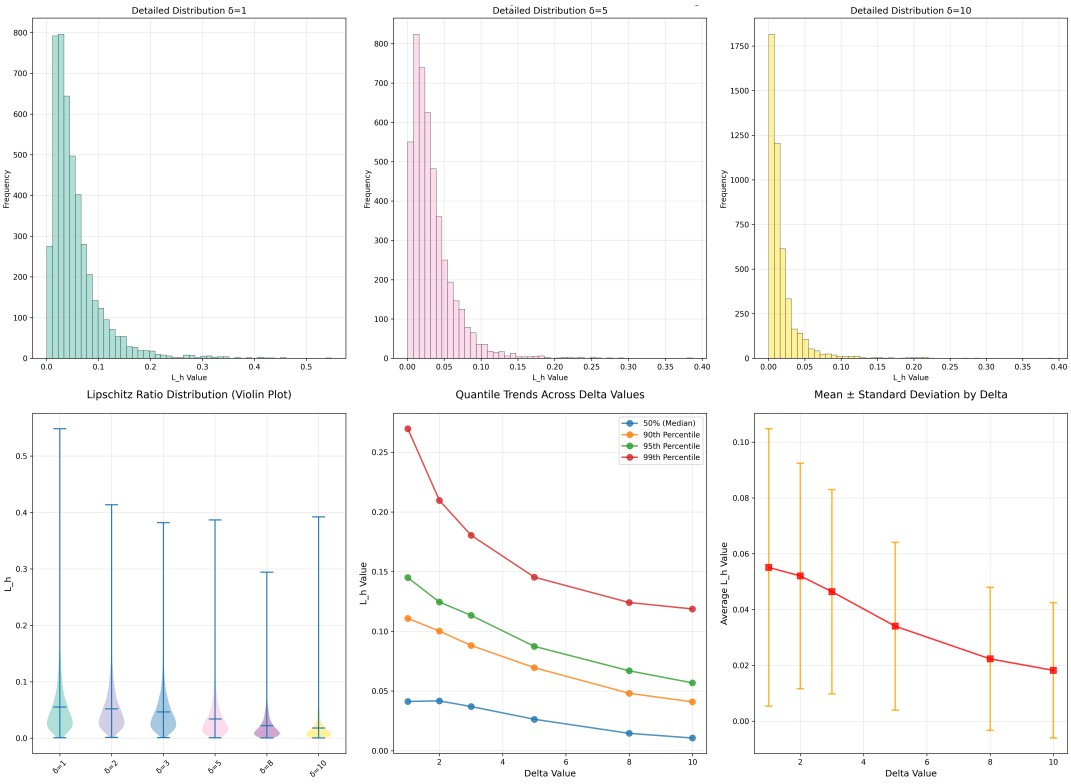

*Figure 6.* Qwen3-Embedding-4B: Local Lipschitz analysis across $\delta$.

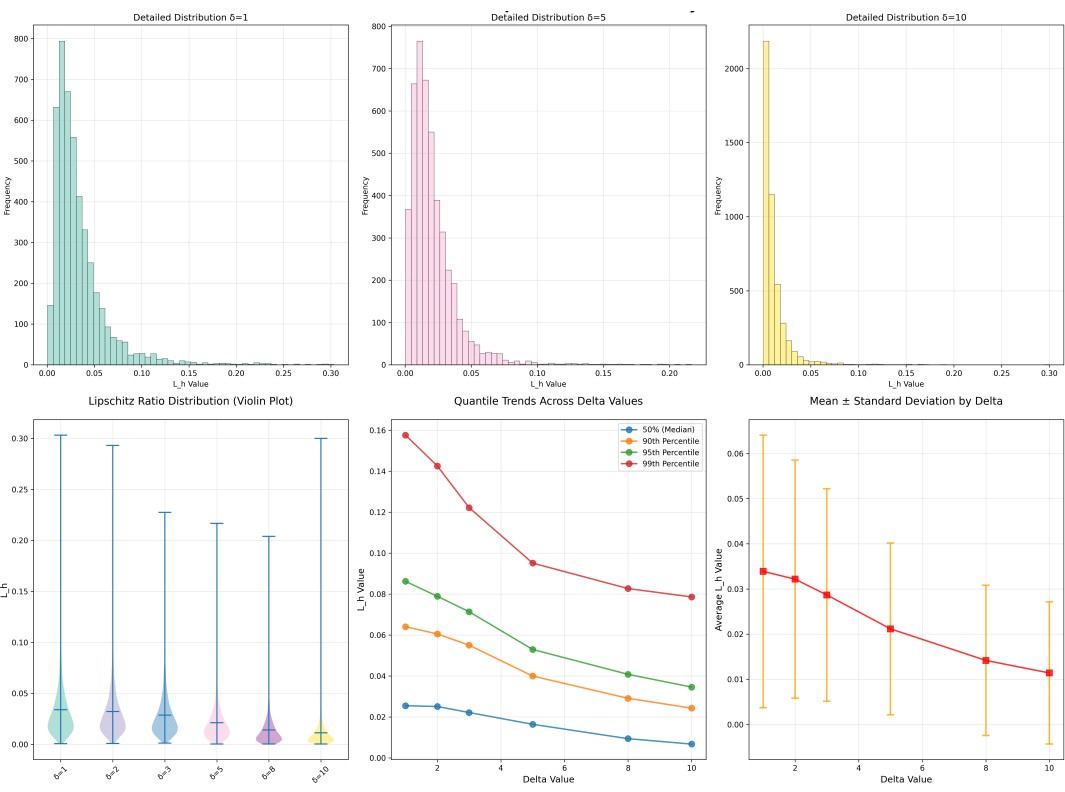

*Figure 7.* Qwen3-Embedding-8B: Local Lipschitz analysis across $\delta$.

**Analysis.** Across radii $\delta \in \{1, 2, 3, 5, 8, 10\}$, the empirical local Lipschitz constant $L_{\text{emp}}(\delta)$ *decreases* as $\delta$ grows, indicating smoother behavior for larger neighborhoods (finite-difference estimates are dominated by local saturation and curvature). As shown in Table 7 and Figure 5, 6, 7, scaling the encoder from 0.6B→4B→8B consistently *reduces* $L_{\text{emp}}$ at all quantiles: for example at $\delta=5$, P95 drops from $0.1107$ (0.6B) to $0.0874$ (4B) to $0.0530$ (8B). Tail risk also shrinks (P99 and Max), with occasional outliers on 0.6B (e.g., $\delta=8$) suggesting numeric spikes; hence, for theoretical bounds we recommend using the *high-probability* constant $L_{\text{loc}} = \text{P95}$ at $\delta \in [3, 5]$ as a robust plug-in for the local bound.

### A.3. Why Concatenate TWO Representation Paths?

Although feature concatenation introduces an additional error source compared to using a single feature vector, which appears to increase the overall error term in the model, we provide an information-theoretic analysis showing that feature concatenation leads to more stable results. Model the two paths as noisy channels:

$$g(x) = s + \eta_g, \qquad h(y) = s + \eta_h,$$

with zero-mean, finite-covariance noises conditionally independent given $s$: $p(\eta_g, \eta_h \mid s) = p(\eta_g \mid s)\, p(\eta_h \mid s)$, where $\eta_g$ and $\eta_h$ are additive noises in the two channels, assumed zero-mean with finite covariance and conditionally independent given $s$. $\sigma_{s,k}^2 = \text{Var}(s_k)$ is the variance of the $k$-th coordinate of the latent semantic vector $s$.

Define the information gain of concatenation:

$$\Delta I \;=\; I\big(s;\, [g(x), h(y)]\big) \;-\; I\big(s;\, g(x)\big),$$

where $I(\cdot\,;\,\cdot)$ and $H(\cdot \mid \cdot)$ denote Shannon mutual information and conditional entropy, respectively. By the chain rule and conditional independence,

$$
\begin{aligned}
I\big(s;\, [g, h]\big) &= I(s; g) + I(s; h \mid g) \\
&= I(s; g) + H(s \mid g) - H(s \mid g, h) \\
&\geq I(s; g), \tag{24}
\end{aligned}
$$

with equality iff $s \perp\!\!\!\perp h \mid g$ (i.e., $h(y)$ provides no additional semantic information beyond what is already contained in $g(x)$). In cross-lingual settings, translation noise $\eta_h$ and anchoring noise $\eta_g$ are complementary, hence $I(s; h \mid g) > 0$ and $\Delta I > 0$. Therfore, if $\text{Var}(\eta_{g,k}) \to \infty$ on some dimension $k$ (e.g., severe LRL ambiguity), then

$$I(s; g) \;\leq\; \tfrac{1}{2} \log\!\Big(1 + \tfrac{\sigma_{s,k}^2}{\text{Var}(\eta_{g,k})}\Big) \to 0.$$

The quantity in parentheses is the per-coordinate signal-to-noise ratio (SNR), defined as $\text{SNR}_k = \sigma_{s,k}^2 / \text{Var}(\eta_{g,k})$. For the additive channel $G_k = s_k + \eta_{g,k}$, the classical Gaussian-channel bound implies $I(s_k; G_k) \leq \tfrac{1}{2} \log(1 + \text{SNR}_k)$. while a stable English path $(\text{Var}(\eta_{h,k}) < \infty)$ yields a strictly positive lower bound for $\Delta I$ on that dimension. Hence the concatenation $\mathbf{z} = [g(x); h(y)]$ overcomes single-path bottlenecks, and the information gain offsets the apparent worst-case bound increase.

## B. Dataset

**Dataset release.** We release a de-identified cross-lingual e-commerce retrieval dataset, **LazRetrieval**, and its pretraining-scale companion **LazRetrieval-mega**. Our dataset is derived from anonymized user click logs from a certain e-commerce platform. We constructed sample pairs based on user search queries and the titles of the products clicked during those searches, thereby creating the retrieval dataset. The corpus spans seven languages across Southeast and South Asia: Vietnamese (vi), Thai (th), Indonesian (id), Malay (ms), Urdu (ur), Bengali (bn), and Filipino/Tagalog (ph). **LazRetrieval** contains $10\,\text{k}$ examples per language, while **LazRetrieval-mega** contains $1{,}000\,\text{k}$ per language. Unless otherwise specified, our experiments use *LazRetrieval*; the *mega* version is intended to support large-scale pretraining. We normalize the frequencies.

**Splits and file structure.** We split the data into train and test with a fixed 4:1 ratio. Each split consists of three JSON files:

- `query.json`: de-identified user queries from seven Lazada locales.
- `item.json`: product titles (landing-page headers) to be retrieved as candidate documents.
- `pairs_info.json`: the set of positive query–item pairs (binary relevance).

**Example.** A minimal example from the Bengali (Bangladesh) portion is shown below.

**query**:
```
{
    "ID":"Q1048",
    "nation": "BD",
    "text": "সট চুইংগাম"
}
```

**item**:
```
{
    "nation": "BD",
    "item_id": "C34801",
    "text": "ডুভেই ক্লাসিক পুরুষ ব্রেসলেট গহনা মুকুট কবজ বিলাসবহুল
ম্যাক্রাম জপমালা মহিলাদের জন্য ব্রেসলেট পালসেইরা ম্যাসকুলিনা ফেমিনিনা
উপহার"
},
```

**pairs_info:**
```
{
    "nation": "BD",
    "query_id": "Q1048",
    "query": "সট চুইংগাম",
    "item_id": "C369",
    "item": "Trident cinnamon ক্লেভার সুগার ফ্রি গাম x 14 সফট গাম",
    "rscore": "1.0"
}
```

*Figure 8.* Rendered examples from the Bengali (BN) split of *LazRetrieval*. We render records as images to avoid Unicode rendering issues.

**Dataset analysis.** As shown in Table 8 and Figure 9 The corpus exhibits a pronounced length asymmetry between queries and items: queries are short on average (mean 18.66, median 18), whereas item titles are substantially longer and more dispersed (mean 97.42, std 42.59). This mismatch reflects real-world e-commerce behavior—concise user intents versus verbose product titles—and implies (i) robust handling of extreme-length outliers and (ii) sensitivity to multilingual scripts with different orthographic granularity. Our training objectives (queue-augmented correlation for sentence ranking and listwise soft-nDCG@10 with safe negatives for retrieval) are designed to be stable under such length skew, while the anchoring mechanism in LiRA mitigates representation drift caused by noisy or unusually long inputs.

## C. Experimental Details

Unless stated otherwise, inputs are tokenized with `max_length=512`, `padding=true`, and truncation enabled. For consistency across backbones, document- and query-side representations are pooled before similarity scoring. Training updates three learnable components: the multilingual encoder (e.g., mT5-XL encoder), the cross-space `Adaptor`, and

*Table 8.* Descriptive statistics of *LazRetrieval* (train+test). Lengths are measured as raw string lengths; counts denote unique entries.

| Field | Max | Min | Mean | Median | Std | Count |
|---|---|---|---|---|---|---|
| Query | 248 | 2 | 18.65 | 18.00 | 9.37 | 50,000 |
| Item | 255 | 6 | 97.42 | 95.00 | 42.59 | 50,000 |

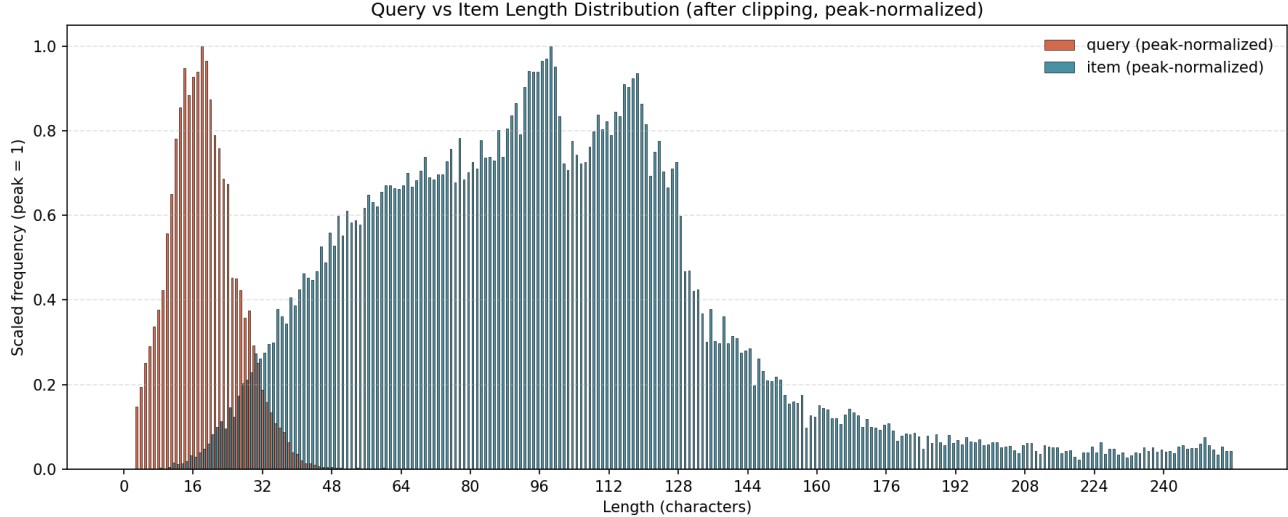

*Figure 9.* Length's distribution of *LazRetrieval*.

the `Actor` selector. All are optimized with Adam and gradient clipping at 1.0. Both single-process and DDP training are supported; checkpointing and logging frequencies are configured per task (Tables. 9–10).

### C.1. Arca Translation Selections-Equipped LiRA-Max

As shown in Figure 10. On MLQARETRIEVAL, MGSM, and BELEBELERETRIEVAL, `Qwen3-32B` is the most frequently selected candidate (29.4%, 34.0%, and 33.8%, respectively); together with `DeepSeek-R1-Distill-Qwen-32B` (27.0%, 31.8%, 21.4%), the Qwen-family accounts for the majority of selections (56.4%, 65.8%, and 55.2%). On LAZRETRIEVAL, the distribution is more balanced with `Llama-3.3-70B-Instruct` at 27.2%, `gemma-2-27b-it` at 26.8%, and the Qwen-family totaling 46.0%. Across datasets, ARCA shows a consistent preference toward Qwen-family candidates (Qwen3 or the Qwen-distilled DeepSeek variant), especially on MGSM where they comprise 65.8% of selections. A plausible explanation is architectural alignment and feature-space compatibility with our *frozen evaluator*, which is `Qwen3-32B`. Using the same family for evaluation can introduce a mild inductive bias that favors stylistic and semantic choices characteristic of that architecture. We therefore report these breakdowns to make the potential bias explicit and to encourage future work to cross-check with evaluators from different families.

*Table 9.* Core setup per experiment. All runs use PyTorch + Transformers; distributed training via DDP when `--distributed` is enabled.

| Task | Encoder | LLM Embedding | Pool Size | Batch | Steps | GPUs |
|---|---|---|---|---|---|---|
| STS22 | mT5-XL | Qwen3-Embedding-8B | 8 | 1 | 10 | 4 |
| BelebeleRetrieval | mT5-XL | Qwen3-Embedding-8B | 8 | 1 | 5 | 8 |
| MLQARetrieval | mT5-XL | Qwen3-Embedding-8B | - | 1 | 5 | 8 |
| LazRetrieval | mT5-XL | Qwen3-Embedding-8B | 4 | 1 | 120 | 8 |
| MGSM | mT5-XL | Qwen3-8B | 4 | 2 | - | 1 |
| X-CSQA | mT5-XL | Qwen3-8B | 4 | 2 | - | 1 |

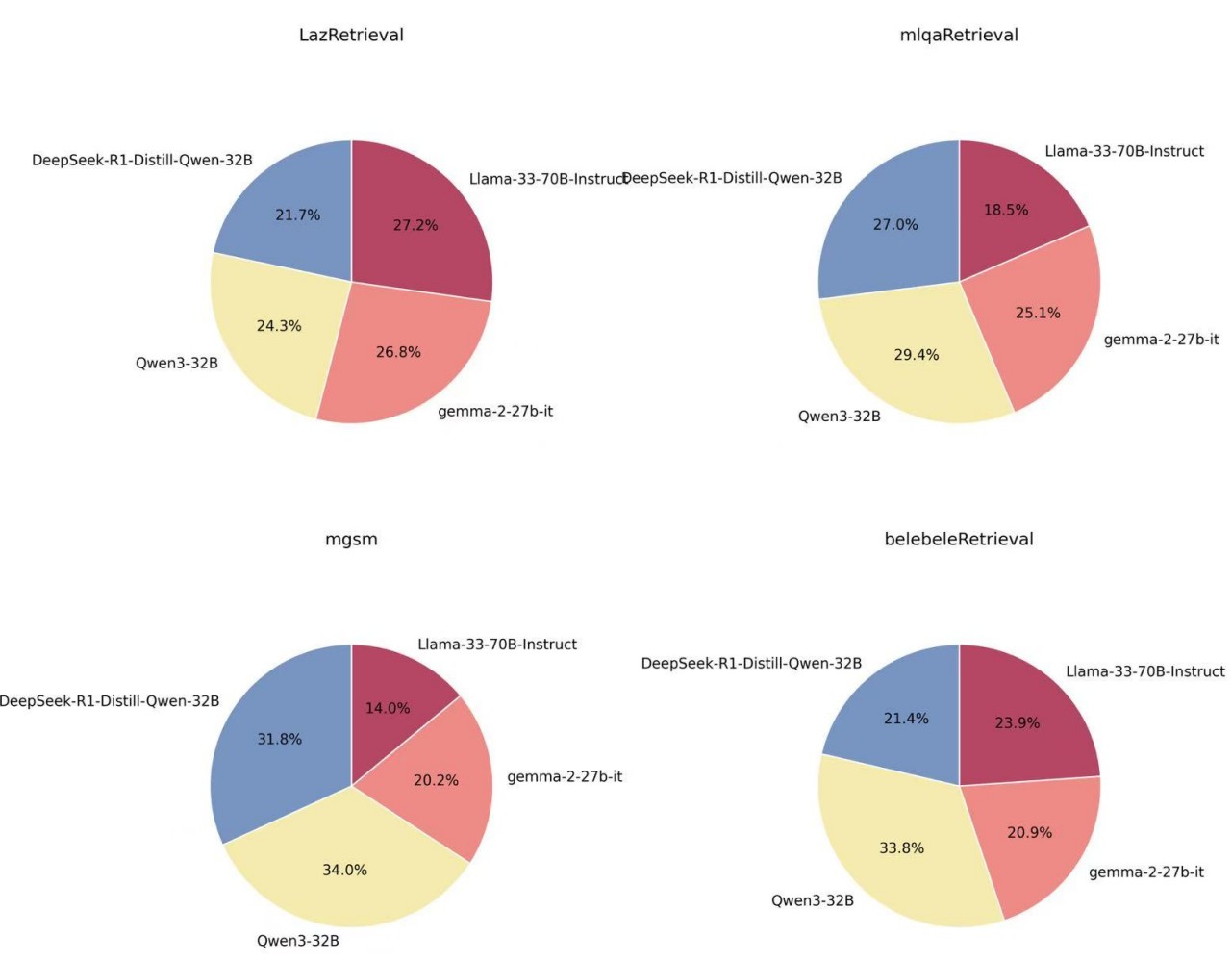

*Figure 10.* Model selection distribution of ARCA across four benchmarks.

### C.2. Prompt Engineer

Our translation-aware training relies on a translator module and a critic module. In practice, however, off-the-shelf LLM translators may produce *non-translation artifacts* (e.g., meta prefixes such as "Here is the translation:", unnecessary politeness, or extra explanations), which introduce avoidable noise into both the translation candidates and the downstream scoring signals. To reduce such artifacts and to make the translation outputs more consistent across languages, we adopt a simple but effective prompt-engineering strategy for both translation and evaluation.

**Translation prompt.** We instruct the model to act as a professional translator and *output only the translated text* without any additional commentary:

> You are a professional translator, proficient in various languages. Please translate the following phrase or sentence into English: '{text}'. Output only the translation results without any other explanations.

**Critic prompt.** To compare multiple translation candidates, the critic evaluates each candidate on three dimensions—semantic fidelity, emotional consistency, and pragmatic tone—and is required to output *strict JSON* to avoid verbose judgments that are hard to parse:

> Please strictly evaluate the following translation on three dimensions:

*Table 10.* Optimization and reward-related hyperparameters. $\alpha, \beta, \gamma$ weight the three translation scores; $\delta$ scales the similarity term. Logging and checkpoint intervals are task-specific.

| Task | Actor LR | Encoder LR | Adaptor LR | $\alpha$ | $\beta$ | $\gamma$ | $\delta$ |
|---|---|---|---|---|---|---|---|
| STS22 | $1 \times 10^{-4}$ | $5 \times 10^{-5}$ | $1 \times 10^{-4}$ | 0.4 | 0.3 | 0.3 | 1.0 |
| BelebeleRetrieval | $1 \times 10^{-4}$ | $5 \times 10^{-5}$ | $1 \times 10^{-4}$ | 0.4 | 0.3 | 0.3 | 1.0 |
| MLQARetrieval | $1 \times 10^{-4}$ | $5 \times 10^{-5}$ | $1 \times 10^{-4}$ | 0.4 | 0.3 | 0.3 | 1.0 |
| LazRetrieval | $1 \times 10^{-4}$ | $5 \times 10^{-5}$ | $1 \times 10^{-4}$ | 0.4 | 0.3 | 0.3 | 1.0 |

- Semantic Fidelity (1–10): Accuracy in conveying original meaning
- Emotional Consistency (1–10): Alignment with original emotional tone
- Pragmatic Tone (1–10): Appropriateness in contextual usage and style

You MUST only output JSON format with keys `'semantic'`, `'emotional'`, `'pragmatic'` and values 1–10, without any other explanations.

Origin text in {`lang`}: '{`sentence`}'

Translation in En: '{`translation`}'

This design serves two purposes: (i) it suppresses extraneous natural-language responses that otherwise contaminate translation candidates and destabilize training, and (ii) it yields structured, dimension-wise scores that can be directly integrated into the actor–critic learning signal without additional heuristics.

## C.3. Bad Case Analysis

Despite strong translation ability, large language models frequently inject small but non-negligible noise into translation outputs, especially when the input is long, contains news-style formatting, or includes imperative phrases and boilerplate (e.g., "Follow us on Telegram"). A common failure mode is *meta-text leakage*: the model prepends or appends non-translation content such as "Sure, here is the translation:", "Here is the translated sentence:", or polite closing statements like "If you need any other help, I'm glad to help you." Although such artifacts are harmless for human readers, they are problematic in our setting because they (i) change the token distribution of the "translation" string, (ii) inject irrelevant stylistic cues that the encoder may latch onto, and (iii) distort critic scoring by mixing translation quality with adherence-to-instruction behavior. We provide a representative example below. Given a long-form input, some LLM translators return a translation *wrapped* with assistant-style meta prefixes and extra explanations, instead of outputting the translation alone:

**Expected translation candidate:**
```
Baku, Azerbaijan, Jan.  1 ...  ``2019 will go down in history as a year of
in-depth reforms ...''
```
**Noisy translation candidate:**
```
Sure, here is the translated sentence:
``Baku, Azerbaijan, January 1 ...  ``2019 will go down in history as a year of
significant reforms ...''  ...''
```

Such meta-text leakage constitutes translation noise under our theoretical framing (semantic drift and formatting artifacts), and it can propagate into both representation learning and the actor–critic reward signal. Concretely, the added preambles (e.g., "Sure, here is . . .") introduce irrelevant tokens and style markers that are absent from genuine translations, making the translation string less comparable across candidates and languages.

Our prompt-engineering strategy mitigates this issue by enforcing *output-only translation* for the translator and *JSON-only scoring* for the critic. Empirically, this reduces the rate of meta-text leakage and yields more uniform translation candidates in style and formatting, leading to a cleaner training signal and more stable translation-aware optimization, especially for long-form inputs with news-style templates.

**Reproducibility notes.** All models use Adam, gradient clipping (1.0), and cosine similarity on L2-normalized vectors. LLM-side token representations are extracted via `get_input_embeddings()`, and `torch.nan_to_num` is applied defensively during scoring.

# D. Model Details

## D.1. About Model

**Multilingual encoder:** mT5-XL (encoder only; decoder frozen). **LLM Evaluator (frozen):** `Qwen/Qwen3-32B` is used only for evaluation/critique; all its parameters are kept frozen. **LaSR (trainable):** the LaSR module is trained end-to-end during our experiments (gradients do not propagate into the frozen evaluator). When needed, token representations are read via `get_input_embeddings()` (no gradient flow).

**Pooling & shapes.** The backbone outputs `last_hidden_state`, which is pooled to a sentence vector. On the LLM side, raw token embeddings are adaptively average-pooled to a fixed temporal length (`pool_size`, e.g., 32 or 4), then flattened for similarity computation.

**Adaptor.** A two-layer MLP maps $\mathbb{R}^{d_{\mathrm{ML}}} \xrightarrow{\text{Linear + ReLU + LayerNorm}} \mathbb{R}^{512} \xrightarrow{\text{Linear}} \mathbb{R}^{d_{\mathrm{LLM}}}$, aligning multilingual features to the LLM embedding space.

**Actor (candidate selector).** For each candidate, the Actor consumes a 4-D feature vector $[\texttt{semantic}, \texttt{emotional}, \texttt{pragmatic}, \texttt{sim}]$ with topology $\mathbb{R}^4 \to 16 \to 1$ (`ReLU+LayerNorm`). A softmax over candidates defines $\pi(a \mid \mathbf{x})$, and REINFORCE is used:

$$\mathcal{L}_{\text{actor}} = -\log \pi(a \mid \mathbf{x}) \cdot R(a),$$

where $R(a) = 0.1\alpha\,\texttt{semantic} + 0.1\beta\,\texttt{emotional} + 0.1\gamma\,\texttt{pragmatic} + \delta \cdot \texttt{sim}$ (weights in Table 10).

**Encoder/adaptor objective.** We maximize cosine similarity between the selected candidate and the aligned query vector by minimizing

$$\mathcal{L}_{\text{enc}} = -\cos(\text{pool}(\text{Adaptor}(\text{Enc}_{\mathbf{ML}}(x))),\ \text{pool}(\text{Emb}_{\mathbf{LLM}}(y))).$$

Three optimizers update `Actor`, the multilingual encoder, and the `Adaptor`; gradient clipping is applied uniformly (1.0).

## D.2. Measuring $\epsilon_1$ and $\epsilon_2$

Our theory only assumes: (i) there exists a mapping error between representation vectors, and (ii) machine translation may introduce noise. Both phenomena are widely observed in cross-lingual alignment and MT-based transfer settings. We introduce an "ideal" representation vector $\mathbf{z}^\star$ that corresponds to the noisy, error-prone representation $\mathbf{z}$ purely as a mathematical construct, rather than as a strict requirement on real-world conditions. Importantly, the purpose of these assumptions is to *derive and justify* our training objective: Assumptions 1–2 are directly aligned with our loss, and $\mathbf{z}^\star$ serves to emphasize that errors arise along two orthogonal dimensions—*semantic drift* (translation distortion) and *vector mapping mismatch* (representation deviation).

To make these assumptions empirically verifiable, we estimate both error terms using observable proxies during training. For the **representation mapping error** $\epsilon_1$, we track the mismatch between the feature-path representation and the translation-path representation:

$$\hat{\epsilon}_1 := \mathbb{E}_x\Big[\big\|g(x) - h(\hat{y}(x))\big\|_2\Big], \tag{25}$$

where $g(x)$ denotes the feature-path embedding and $h(\hat{y}(x))$ denotes the translation-path embedding produced by the selected translation $\hat{y}(x)$ (e.g., the actor/critic-chosen candidate). In addition, we report the cosine similarity $\cos(\mathbf{z}, \mathbf{z}^\star)$ as a normalized proxy to visualize the contraction of representation deviation across epochs (higher is better).

For the **translation distortion** $\epsilon_2$, since the ideal translation $y^\star$ is not observable, we use a semantic-divergence proxy that captures the degree of drift introduced by translation:

$$\hat{\epsilon}_2 := \mathbb{E}_x\Big[1 - \cos\big(\text{Emb}(x), \text{Emb}(\hat{y}(x))\big)\Big], \tag{26}$$

In Eq. (26), we instantiate $\text{Emb}(\cdot)$ using the frozen encoder states of $f_{\text{llm}}$ (i.e., the LLM representations before decoding). Concretely, given an input text $u$ (either the source sentence $x$ or the selected translation $\hat{y}(x)$), we obtain token-level hidden

states $\mathbf{H}_u = \mathrm{Enc}_{f_{\mathrm{llm}}}(u) \in \mathbb{R}^{T \times D}$, and compute a sentence-level embedding by masked mean pooling:

$$E_{f_{\mathrm{llm}}}(u) \;=\; \frac{1}{\sum_t m_t} \sum_{t=1}^{T} m_t\, \mathbf{H}_u[t], \tag{27}$$

where $m_t \in \{0, 1\}$ is the attention mask. We keep $f_{\mathrm{llm}}$ *frozen* and use a fixed prompting/formatting template for all inputs in this appendix to ensure $E_{f_{\mathrm{llm}}}(\cdot)$ is a stable diagnostic signal.

Accordingly, our translation-distortion proxy is measured as

$$\hat{\epsilon}_2 \;:=\; \mathbb{E}_x \Big[ 1 - \cos\big(E_{f_{\mathrm{llm}}}(x),\, E_{f_{\mathrm{llm}}}(\hat{y}(x))\big) \Big]. \tag{28}$$

Lower $\hat{\epsilon}_2$ indicates better semantic faithfulness and reduced translation noise (as measured in the representation space of $f_{\mathrm{llm}}$).

Table 11 reports the evolution of the similarity between $\mathbf{z}$ and $\mathbf{z}^\star$ (cosine similarity proxy) and the corresponding training loss across epochs on three representative Arca training tasks. We observe a consistent increase in similarity and a monotonic decrease in loss, which is consistent with the theoretical interpretation that the representation deviation contracts as training proceeds. Table 12 summarizes the measured diagnostics $\hat{\epsilon}_1$ and $\hat{\epsilon}_2$ under different training settings.

*Table 11.* Similarity between $\mathbf{z}$ and $\mathbf{z}^\star$ (cosine similarity proxy) and training loss across epochs (averaged over three Arca training tasks). Higher similarity and lower loss indicate improved anchoring and reduced deviation.

| Epoch | 1 | 2 | 3 | 4 | 5 | 6 | 7 | 8 | 9 | 10 |
|---|---|---|---|---|---|---|---|---|---|---|
| **Sim.** | 0.01 | 0.14 | 0.34 | 0.45 | 0.54 | 0.60 | 0.64 | 0.67 | 0.69 | 0.71 |
| **Loss** | 1.94 | 1.23 | 0.97 | 0.74 | 0.62 | 0.54 | 0.42 | 0.35 | 0.29 | 0.21 |

*Table 12.* Diagnostics for representation mapping error and translation distortion. Lower is better for $\hat{\epsilon}_1$ and $\hat{\epsilon}_2$. Fill in with your measured values (mean $\pm$ std if available).

| Setting | Task | $\hat{\epsilon}_1 \downarrow$ | $\hat{\epsilon}_2 \downarrow$ | Sim. $\uparrow$ | nDCG@10 $\uparrow$ |
|---|---|---|---|---|---|
| Qwen3-Embedding-8B | LazRetrieval | 0.81 | 0.54 | 0.01 | 69.4 |
| +Arca | LazRetrieval | 0.19 | 0.23 | 0.78 | 69.9 |
| +Arca + LaSR | LazRetrieval | 0.19 | 0.23 | 0.78 | 71.1 |

## D.3. A more complex training pipeline?

While LiRA introduces translation-aware training, our pipeline is designed to be *budget-flexible* and *engineering-friendly*. First, translations can be prepared *offline* in advance, which substantially reduces online training overhead. Second, the translator can be replaced by a smaller MT model when compute is constrained. We report resource usage for both training and inference in Table 13 and 14, measured in single A100-80G GPU hours.[1]

Here, `pass@k` indicates that $k$ LLM translators are used along the forward path (i.e., $k$ translation candidates are produced by LLMs); `pass@0` uses only MT models. In practical deployment, one may combine a lightweight MT model with an LLM (or use MT only). Notably, even `pass@0` already outperforms the original Qwen3-Embedding-8B baseline in our experiments, indicating that LiRA provides a principled way to customize computation budgets and select translator capacity according to available resources.

*Table 13.* Training resource usage (single A100-80G GPU hours). `pass@k` denotes using $k$ LLM translators along the forward path.

| Setting | Pass@4 Arca | Pass@4 LaSR | Pass@2 Arca | Pass@2 LaSR | Pass@0 Arca | Pass@0 LaSR |
|---|---|---|---|---|---|---|
| Offline translation | 0.15h | 0.20h | 0.15h | 0.20h | 0.15h | 0.20h |
| Online translation | 1.50h | 2.00h | 0.80h | 1.10h | 0.30h | 0.40h |

---

[1]The reported "GPU hours" measure the end-to-end wall-clock GPU time under our implementation setup.

*Table 14.* Inference resource usage (single A100-80G GPU hours) under different `pass@k` settings.

| Setting | pass@4 | pass@2 | pass@0 |
|---|---|---|---|
| Offline translation | 0.30h | 0.30h | 0.30h |
| Online translation | 3.80h | 2.10h | 0.90h |

**Pass@k.**  We evaluate with a $k$-way budget: *pass@k* is the number of LLM translators used in one LiRA forward pass ($k=0$ uses only MT; $k>0$ follows Table 15). Table 16 aggregates MLQA Retrieval, BelebeleRetrieval, and STS22. Across $k \in \{0, 1, 2, 3, 4\}$, LiRA outperforms the baseline on all three datasets. Both models improve as $k$ increases, with diminishing gains beyond $k=2$ and a small dip at $k=3$ on MLQA. Even at $k=0$, LiRA beats the baseline, showing a tunable budget–quality trade-off without large translators at inference.

*Table 15.* Translator configurations for pass@k (abbreviations: OPUS = OPUS-MT; M2M = m2m100; N600 = nllb-200-600M; N3B = nllb-200-3.3B; DS-R1 = DeepSeek-R1-Distill-Qwen-32B).

| PASS@K | T1 | T2 | T3 | T4 |
|---|---|---|---|---|
| PASS@0 | OPUS | M2M | N600 | N3B |
| PASS@1 | LLAMA | OPUS | M2M | N3B |
| PASS@2 | LLAMA | GEMMA | M2M | N3B |
| PASS@3 | LLAMA | GEMMA | QWEN | N3B |
| PASS@4 | LLAMA | GEMMA | QWEN | DS-R1 |

*Table 16.* Combined pass@k results on three datasets (higher is better).

| METHOD | PASS@K | MLQA-R | BELE-R | STS22 |
|---|---|---|---|---|
| QWEN3-8B | PASS@0 | 79.96 | 82.27 | 69.64 |
|  | PASS@1 | 80.41 | 83.54 | 70.01 |
|  | PASS@2 | 80.45 | 83.97 | 70.72 |
|  | PASS@3 | 80.79 | 84.55 | 71.32 |
|  | PASS@4 | 81.13 | 85.94 | 71.64 |
| WITH LIRA | PASS@0 | 81.15 | 86.00 | 73.01 |
|  | PASS@1 | 81.56 | 86.15 | 73.54 |
|  | PASS@2 | 81.79 | 86.67 | 74.11 |
|  | PASS@3 | 81.53 | 86.69 | 74.39 |
|  | PASS@4 | 82.01 | 87.03 | 75.00 |

## D.4. Effect of Naive Parameter Scaling

To examine whether LiRA's improvements simply come from increasing the number of involved pretrained parameters, we compare LiRA with a naive dual-path combination that directly uses the multilingual and English representation paths without Arca training, critic-guided selection, or LaSR consistency optimization. This setting introduces additional auxiliary models but does not explicitly align the two representation spaces. As shown in Table 17, the naive combination does not improve over the corresponding Qwen3 backbone and slightly underperforms it across languages. This indicates that simply adding pretrained components is insufficient; the gains of LiRA mainly come from explicit anchoring, translation selection, and cross-lingual consistency objectives.

## D.5. Objective

**Ranking objective.**  To improve the statistical stability of correlation targets (Pearson / Spearman) under small batches, we maintain a FIFO history queue of length at most $K$. At each optimization step, we concatenate the *history* (prediction–label pairs) to the current batch and compute the correlation losses jointly; the history is treated as a constant via stop-gradient and never contributes gradients. Let the current batch size be $B$, the number of valid history items be $m \le K$, and the total after concatenation be $N = m + B$. Denote the predicted similarities by $\mathbf{p} = (p_1, \ldots, p_B)^\top$ where

$$p_i = \cos(\mathbf{q}_i, \mathbf{d}_i) = \mathbf{q}_i^\top \mathbf{d}_i \in [-1, 1] \tag{29}$$

*Table 17.* Effect of naive parameter scaling on LazRetrieval. The naive dual-path variant adds auxiliary pretrained components but removes LiRA training objectives.

| METHOD | PARAMS | BN | ID | MS | UR | TH | PH | VI | AVG. |
|---|---|---|---|---|---|---|---|---|---|
| QWEN3-E-0.6B | 0.6B | 38.36 | 63.95 | 62.37 | 40.73 | 55.28 | 74.38 | 59.46 | 56.31 |
| W/O LiRA TRAINING | 8.5B | 37.59 | 62.53 | 61.71 | 40.18 | 54.62 | 73.79 | 58.91 | 55.62 |
| W/ LiRA-LARGE | 8.5B | 48.60 | 74.43 | 71.26 | 49.84 | 66.39 | 83.90 | 70.67 | 66.44 |
| QWEN3-E-4B | 4B | 49.81 | 68.92 | 71.45 | 49.30 | 73.39 | 78.47 | 68.31 | 65.66 |
| W/O LiRA TRAINING | 11.9B | 49.24 | 68.31 | 70.88 | 48.63 | 72.71 | 77.92 | 67.74 | 65.06 |
| W/ LiRA-LARGE | 11.9B | 56.24 | 75.29 | 77.03 | 55.69 | 77.72 | 84.81 | 74.92 | 71.67 |
| QWEN3-E-8B | 8B | 50.59 | 76.01 | 73.36 | 50.37 | 69.67 | 84.78 | 71.56 | 68.05 |
| W/O LiRA TRAINING | 15.9B | 49.98 | 75.43 | 72.79 | 49.74 | 69.09 | 84.16 | 70.93 | 67.45 |
| W/ LiRA-LARGE | 15.9B | 57.70 | 77.33 | 77.93 | 56.67 | 78.52 | 86.03 | 75.86 | 72.86 |

(the two vectors are $L_2$-normalized sentence embeddings), and the gold scores by $\mathbf{t} = (t_1, \ldots, t_B)^\top$ (e.g., STS22 annotations). The detached history buffers are $\mathbf{p}^{\text{hist}} \in \mathbb{R}^m$ and $\mathbf{t}^{\text{hist}} \in \mathbb{R}^m$. We concatenate

$$\tilde{\mathbf{p}} = \begin{bmatrix} \mathbf{p}^{\text{hist}} \\ \mathbf{p} \end{bmatrix}, \quad \tilde{\mathbf{t}} = \begin{bmatrix} \mathbf{t}^{\text{hist}} \\ \mathbf{t} \end{bmatrix} \in \mathbb{R}^N. \tag{30}$$

If $N < N_{\min}$ (warm-up threshold), we skip the update. *(1) Pearson correlation:*

$$r(\mathbf{a}, \mathbf{b}) = \frac{\frac{1}{N} \sum_{i=1}^N (a_i - \bar{a})(b_i - \bar{b})}{\sqrt{\frac{1}{N} \sum_{i=1}^N (a_i - \bar{a})^2 + \varepsilon} \sqrt{\frac{1}{N} \sum_{i=1}^N (b_i - \bar{b})^2 + \varepsilon}}, \quad \varepsilon = 10^{-8}, \tag{31}$$

applied to $(\tilde{\mathbf{p}}, \tilde{\mathbf{t}})$. *(2) Soft-Spearman (differentiable rank correlation):* first compute soft ranks $R_i$ for $\tilde{\mathbf{p}}$ with temperature $\tau > 0$,

$$R_i(\tilde{\mathbf{p}}; \tau) = 1 + \sum_{j=1}^N \sigma\left(\frac{\tilde{p}_i - \tilde{p}_j}{\tau}\right), \qquad \sigma(x) = \frac{1}{1 + e^{-x}}. \tag{32}$$

The label ranks $\rho_i(\tilde{\mathbf{t}})$ use average ties (the standard statistical convention). Define Soft-Spearman as *Pearson on ranks*:

$$r_s(\tilde{\mathbf{p}}, \tilde{\mathbf{t}}) = r\Big( \mathbf{R}(\tilde{\mathbf{p}}; \tau), \boldsymbol{\rho}(\tilde{\mathbf{t}}) \Big). \tag{33}$$

Note that as $\tau \to 0$, $\mathbf{R}(\cdot; \tau)$ approaches discrete ranks; smaller $\tau$ sharpens sorting but increases gradient variance. *(3) Combined loss (as implemented):* for $\alpha \in [0, 1]$,

$$\mathcal{L}_{\text{CorrQ}} = \alpha \left(1 - r(\tilde{\mathbf{p}}, \tilde{\mathbf{t}})\right) + (1 - \alpha)\left(1 - r_s(\tilde{\mathbf{p}}, \tilde{\mathbf{t}})\right). \tag{34}$$

In practice, $\mathbf{p}^{\text{hist}}$ is passed via `detach()`, so gradients of $\mathcal{L}_{\text{CorrQ}}$ flow only through the current $\mathbf{p}$. The queue is updated in FIFO fashion to keep at most $K$ entries (module `corr_queue`). The warm-up threshold $N_{\min}$ (module `corr_min_effective`) enqueues without backprop when data are insufficient, stabilizing early training.

**Retrieval objective.** *Problem setup.* For each query $q$, build a candidate set $\mathcal{C} = \{d_0, \ldots, d_{C-1}\}$ where $d_0$ is the positive and the rest are online hard negatives (in-batch or mined from a same-language index). With qrels, obtain binary r elevance $y_i \in \{0, 1\}$. Use inner-product/cosine scores

$$s_i = \langle \hat{q}, \hat{d}_i \rangle, \qquad \hat{q} = \frac{q}{\|q\|}, \hat{d}_i = \frac{d_i}{\|d_i\|}. \tag{35}$$

*Differentiable ranks.* Use descending soft ranks (SoftRank) with temperature $\tau$:

$$r_i = 1 + \sum_{j \neq i} \sigma\left(\frac{s_j - s_i}{\tau}\right), \qquad \sigma(\cdot) \text{ is the sigmoid.} \tag{36}$$

Larger $s_i$ yields $r_i$ closer to $1$. *Soft nDCG@k.* Define the discount and soft top-$k$ mask as

$$\text{disc}_i = \frac{1}{\log_2(1 + r_i)}, \qquad m_i = \sigma\left(\frac{k + \frac{1}{2} - r_i}{\tau_k}\right). \tag{37}$$

With gains $g_i = 2^{y_i} - 1$,

$$\text{DCG} = \sum_i g_i \, \text{disc}_i \, m_i, \qquad \text{IDCG} = \sum_{t=1}^{k}(2^{y_t^{\downarrow}} - 1)\frac{1}{\log_2(1 + t)}, \qquad \text{nDCG@}k = \frac{\text{DCG}}{\max(\text{IDCG}, \varepsilon)}. \tag{38}$$

The base loss is

$$\mathcal{L}_{\text{ndcg}} = 1 - \text{nDCG@}k. \tag{39}$$

*Safe negatives (near-negative safety gate).* To avoid treating unlabeled relevant items as negatives, set

$$\theta = s_+ - \delta, \quad s_+ = s_0, \tag{40}$$

and identify near negatives $\{i : \ y_i = 0, \ s_i \geq \theta\}$. Apply continuous down-weighting

$$w_i = \begin{cases} \sigma\left(\dfrac{\theta - s_i}{\beta}\right), & y_i = 0 \\ 1, & y_i = 1 \end{cases} \tag{41}$$

and update $\text{disc}_i \leftarrow w_i \, \text{disc}_i$ (or drop near negatives as a more aggressive variant). *Stability terms.* To mitigate collapse and evaluation jitter, add two lightweight regularizers: (i) Top-1 hinge to enforce a margin between the positive and the hardest negative,

$$\mathcal{L}_{\text{hinge}} = \max\left(0, \ \gamma + \max_{y_i=0} s_i - s_+\right); \tag{42}$$

(ii) Mean/variance regularization to control score centering and energy,

$$\mathcal{L}_{\text{mv}} = (\bar{s})^2 + \big|\, \text{Var}(s) - \nu\,\big|, \qquad \bar{s} = \tfrac{1}{C}\sum_i s_i. \tag{43}$$

*Final objective.* The per-sample objective is

$$\mathcal{L} = \mathcal{L}_{\text{ndcg}} + \lambda_h \, \mathcal{L}_{\text{hinge}} + \lambda_r \, \mathcal{L}_{\text{mv}}, \tag{44}$$

and the training loss is the batch mean. In practice we keep only in-batch negatives and take the $M$ hardest (top-$M$) to control complexity; when using external candidates (e.g., same-language queues/indices), we re-score with the current model and then take top-$M$ to reduce stale hard-negative artifacts. *Implementation notes.* We set $\tau \in [0.05, 0.2]$, $\tau_k \approx 0.5$, $\delta \in [0.1, 0.3]$, $\beta \in [0.01, 0.05]$, $\gamma \approx 0.05$, $\nu \approx 0.15$, normalize scores, apply global gradient clipping, and use EMA for evaluation. This objective directly maximizes differentiable nDCG@k while safe negatives and steady-state regularization prevent periodic collapse due to score-field saturation.

## E. Limitations

LiRA has several limitations. First, it mainly targets mid- and low-resource languages that can still be recognized by existing pretrained encoders and translation models; for extremely low-resource or endangered languages with very limited digital presence, additional language-specific adaptation may be required. We acknowledge that "low-resource" is a multidimensional concept and should not be defined by corpus size alone. LiRA mainly targets mid- and low-resource languages that can still be recognized by existing encoders and translation models. For extremely low-resource or endangered languages with limited digital presence, weak MT support, or poor tokenizer coverage, additional language-specific adaptation may be required before applying LiRA. In domain-specific scenarios such as e-commerce retrieval, languages like Urdu and Bengali also remain under-supported due to the lack of high-quality annotated data, which motivates the release of LazRetrieval. Meanwhile, although translations can be pre-computed offline and reused, candidate generation and critic-based selection still introduce extra preprocessing cost, so our "lightweight" claim mainly refers to trainable adaptation and post-preprocessing inference. Finally, our reasoning experiments focus on final-answer accuracy rather than explicit multilingual chain-of-thought generation, which we leave for future work.

