# OpenReview forum: "Toward Robust Multilingual Adaptation of LLMs for Low-Resource Languages"
_ICML.cc/2026/Conference — ICML 2026 regular_

### Official Review · Reviewer_dE2m · 2026-03-11

**Soundness:** 4
**Presentation:** 3
**Significance:** 3
**Originality:** 4
**Overall Recommendation:** 5
**Confidence:** 3

**Summary:**

The paper presents a FT framework (LiRA) that optimizes representation stability and cross-lingual semantic consistency, with the aim of improving English-centric LLMs performance on low-resource languages, particularly leveraging the source LLM’s reasoning capabilities.

LiRA integrates two components: one which reduces semantic drift by aligning multilingual representations with English via critic–actor interaction and feature anchoring; and another one that fuses multilingual and English embeddings with queue-based objectives to support robust retrieval and reasoning in low-resource settings.

The authors also offer a product retrieval dataset (LazRetrieval) covering 5 South-east Asian and 2 South Asian mid- and low-resource languages for further use.

-Theoretical results show that under controlled anchoring error and translation-induced bias, LiRA does what it aims to do. However, increase in model param size are considerable for the reported gains.

**Compliance With Llm Reviewing Policy:**

Affirmed.

**Final Justification:**

The additional clarifications on the points mentioned will definitely help round out the paper and improve its overall quality. Given that these aspects are now resolved, I am happy to maintain my score of 5

**Key Questions For Authors:**

See questions under “Weaknesses”. Clarifying these may make me go up in the “Overall Recommendation”.

**Limitations:**

No, please clarify potential limitations of the work. Especially the increase in model param size mentioned above.

**Strengths And Weaknesses:**

Strenghts:
- The proposed framework (LiRA) provides formal guarantees on the completeness and stability of the learned representations, something other existing FT methods do not always do.

- While reading the paper I felt I wanted to share it with my colleagues, but this shows the benefits of having this paper out in the open.

- Very well-written paper with strong justifications and explanations. Very good in including useful supplementary experiments, ablation study, and further details in the Appendices.

Weaknesses and minor comments:
- It is unclear to me how translation-induced noise is generated or validated as “noise”. Is it assumed that all translations generate noise? Is it done by using multiple models as translators (as seen in Fig 2)? I see some potential superficial mention to the fact in 4.1.2; but nothing else. Apologies if I missed further explanations and understandings in-text.

- Rather than fighting against existing internal distribution, LiRA aims at leveraging them for better multilingual performance. You mention that this is done with a lightweight FT. However, in Table 1, resulting models with LiRA seem to be considerably larger that their source models. For instance, QWEN3-E-0.6B with LiRA-Large goes from 0.6B parameters to 8.5B; all this for a 10 point average increase. While 10 points are fantastic, the huge size jump minimizes the usability of the model down the line. Am I missing something here?

- I missed a discussion of results and a limitations section, though I appreciate all the details provided. Maybe add this for the camera-ready.

---

> ### Author Rebuttal · Authors · 2026-03-30
>
> Thank you so much for your incredibly thoughtful and encouraging review! We're truly honored that you found the paper worth sharing with colleagues — that means a lot to us. We really appreciate your recognition of LiRA's theoretical guarantees, clarity, and supplementary experiments. Your questions about translation-induced noise modeling and the parameter-size trade-off are spot-on and very helpful; we've clarified these points carefully below and will add a dedicated limitations discussion in the camera-ready version. Thanks again for your strong support and constructive feedback!
>
> **About Translation Noise. (W1 part I)** It is generally accepted that an interesting and widely recognized phenomenon is that if a text is translated back and forth multiple times (for example, translating a Shakespearean poem into Chinese and then back into English, or translating certain chapters of Harry Potter into Spanish and then back into English), the content of the text will become completely unrecognizable. This means that translation is always accompanied by noise, and such noise accumulates. Furthermore, several studies [1][2][3] have demonstrated this point.
>
> **Is it done by using multiple models as translators? (W1 part II)** We designed Arca to minimize translation noise as much as possible and developed mathematical models for both translation noise and semantic space noise. During the translation process, our actor uses the method shown in Fig.2 to select the translation candidate with the lowest noise. Furthermore, the actor employs reinforcement learning to filter out non-traditional noise present in the translation (Appendix C.3, Lines 1134–1140, Page 21) to minimize both types of noise identified in our theoretical model.
>
> **About Model parameters. (W2)** A significant portion of the parameters in our method includes those of the translation model; however, since the translation of data can be prepared offline prior to training or inference, from another perspective, the parameters of the translation model can be excluded from the total number of parameters in our model. For example, the translation model in our LiRA-Base has approximately 4.3 billion parameters, while the Critic model has 1.7 billion parameters. Since these datasets can be prepared offline, we consider the training and inference of our method to remain lightweight. Furthermore, by pre-preparing candidate translation texts for various large datasets in advance and making them open source, our method conserves computational resources for future research; this preparation process is typically performed once and can be reused multiple times.
>
> **About Limitation. (W3)** The limitation of our approach is that if we encounter languages with extremely limited data (such as certain tribal languages in Africa), which large language models cannot recognize or encode about, our method will produce corresponding inaccuracies.
> To address this, we can replace the pretrained backbone with a model capable of encoding these languages—for instance, by first fine-tuning the pretrained model on available collected corpora to establish basic recognition. Our method can then transfer capabilities learned from high-resource languages to these low-resource targets. This adaptability constitutes the core value of our approach.
> We will incorporate this discussion into our limitations and future work.
>
> [1] Translation Artifacts in Cross-lingual Transfer Learning.
>
> [2] On the Evaluation Practices in Multilingual NLP: Can Machine Translation Offer an Alternative to Human Translations?
>
> [3] Is it Good Data for Multilingual Instruction Tuning or Just Good Multilingual Evaluation Data?

---

> > ### Author Rebuttal · Reviewer_dE2m · 2026-04-07
> >
> > Many thanks for all the explanations and for agreeing to add a limitations section. Also, I feel that clarifying some (of all) of the aspects I mentioned would help round up your paper. Now I feel even more confident on my overall 5.

---

### Official Review · Reviewer_jsjx · 2026-03-12

**Soundness:** 2
**Presentation:** 4
**Significance:** 3
**Originality:** 3
**Overall Recommendation:** 6
**Confidence:** 3

**Summary:**

The authors propose LiRA, a framework to improve low-resource multilingual retrieval and reasoning by anchoring non-English inputs into an English semantic space with theoretical formulation supported by empirical results.

**Compliance With Llm Reviewing Policy:**

Affirmed.

**Final Justification:**

I recommend that this work be accepted. The study proposes improving low-resource multilingual retrieval and reasoning by anchoring non-English inputs into an English semantic space. The authors model both anchoring error and translation distortion error with a strong theoretical formulation. The approach is sound, original, clearly presented, and highly relevant.
Throughout the rebuttal, the authors have addressed all concerns, reinforcing my prior assessment,  particularly regarding the carbon footprint and the costs of model training and inference, as well as how the approach can be adapted to even lower-resource languages, with a commitment to include these details in the appendix. Therefore, I recommend acceptance of this outstanding work.

**Key Questions For Authors:**

1. Could you please elaborate on the hyperparameter designs of Arca? Would a simpler selection process be able to replace the Actor Model? Have you experimented on these?
2. Could you please elaborate more on how the LazRetrieval dataset was constructed?
3. How would the results differ in the case where a certain component of Arca fail due to the scarcity of the language used?
4. Could you please elaborate more on the costs of using your method (training and inference).

**Limitations:**

yes

**Strengths And Weaknesses:**

Strengths:
- Clear motivation and novelty: LiRa decomposition into Arca and LaSR is quite novel, with explicit modelling of the anchoring error and the translation distortion error. The results also seem quite robust and consistent.  Experiments cover public benchmarks as well as evaluation on a dataset that will be released by the authors.
- The formalization is clearly written and easy to follow. The authors presented an outstanding set of additional material in the appendix (which includes proofs and experimental details such as costs of using the method).  The method also does not rely on parallel data.
- Conceptually, the paper pushes the field towards explicitly modeling and controlling translation noise and representation mapping error, rather than treating cross-lingual adaptation as a black-box fine-tuning problem.


Weaknesses:
- The training pipeline is quite complex: multiple translators, a  LLM critic, an embedding critic, and an actor trained with REINFORCE. This raises concerns about tuning/sensitivity to many hyperparameters, as well as carbon footprint. We are aware that the method is budget-flexible, however, it is important to include part of such discussion in the main text.
- The authors propose to translate low-resource languages using several translators. There was no discussion about extremely low-resource data that is not well supported by the models.
- The authors are planning to release the LazRetrieval dataset and describe it in more detail in the appendix. It is important to include more information about how this dataset was constructed (for example, the  data collection, de-identification).

---

> ### Author Rebuttal · Authors · 2026-03-30
>
> Thank you so much for your thoughtful review and positive assessment! We truly appreciate your time in evaluating our work, and we're delighted that you found LiRA's motivation, theoretical formulation, and novelty compelling. Your constructive feedback on training complexity, dataset details, and low-resource considerations is incredibly helpful — we've addressed each point carefully below, and we'll make sure to incorporate your suggestions into the final version. Thanks again for your support!
>
> **About Hyperparameter (W1 & Q1).** Hyperparameter Sensitivity and Simpler Selection Methods: We experimented with deterministic selection (e.g., greedy selection, which always chooses the translation with the highest original review score) or random selection. In our initial experiments, greedy selection led to unexpected program crashes, so we designed a very simple actor neural network for pattern selection to ensure model stability—a common setup in actor-critic reinforcement learning algorithms. However, as shown in our ablation experiments (removing the LLM Critic or embedding Critic), random selection reduces model performance.
>
> **About Model Cost or Carbon Footprint (W1).** A significant portion of the parameters in our method consists of the translation model’s parameters; however, since the translation of data can be prepared offline before training or inference, from another perspective, the translation model’s parameters can be excluded from our total model parameters. For example, the translation model in our LiRA-Base has approximately 4.3 billion parameters, while the Critic model has 1.7 billion parameters. Since these can be prepared offline, we consider the training and inference of our method to remain lightweight. Furthermore, our method can pre-prepare candidate translations for various large datasets and making them open source, thereby freeing up more computational resources for future research. This reduces potential carbon emissions, as the preparation process is typically performed once and can be reused multiple times.
>
> **Discussion About Extremely Low-resource Data. (W2, Q3)** First, as stated in main contributions, our goal is to transfer English LLM capabilities to mid/low-resource languages. Accordingly, we evaluated across all resource levels (e.g.,  PK-Urdu [1], BD-Bengali in LazRetrieval and Bn-Bengali, Sw-Swahili in MGSM [2] are considered low-resource languages in previous studies; Malay, Vietnamese, Filipino, Thai in LazRetrieval are considered mid-resource languages). Second, Our approach does rely on the pretrained model's ability to recognize and encode the target language. We assume that by "extremely low-resource" languages you are referring to those for which even usable data are scarce (e.g., African tribal languages). If pre-trained models cannot recognize or encode these languages due to the scarcity (which means that a certain component of Arca might fail), or if no usable corpora exist, our method's performance is likely to degrade without adaptation. To address this, we can replace the pretrained backbone with a model capable of encoding these languages—for instance, by first fine-tuning the pretrained model on available collected corpora to establish basic recognition. Our method can then transfer capabilities learned from high-resource languages to these low-resource targets. This adaptability constitutes the core value of our approach. We will incorporate this discussion into our limitations and future work.
>
> **How the LazRetrieval dataset was constructed.(W3, Q2)** Our dataset is derived from anonymized user click logs from a certain e-commerce platform. We constructed sample pairs based on user search queries and the titles of the products clicked during those searches, thereby creating the retrieval dataset. And we will add the detailed version about the LazRetrieval.
>
> **The Detailed Cost of Our Method. (Q4)** Using a single A100 GPU and pre-prepared translation data, taking the STS22 dataset as an example—which contains approximately 8,000 data points—training the Arca module takes about 2 minutes per epoch, with approximately 5,000 MB of VRAM usage. Training the LaSR module takes about 9 minutes per epoch, with approximately 48,000 MB of VRAM usage. The time and memory consumption per data point during inference are roughly the same as during training (since the translation data is prepared offline and does not need to be recalculated each time). We will include more specific experimental details in the appendix.
>
> [1] Building and Evaluating a High Quality Parallel Corpus for English Urdu Low Resource Machine Translation.
>
> [2] MindMerger: Efficiently Boosting LLM Reasoning in non-English Languages.

---

> > ### Author Rebuttal · Reviewer_jsjx · 2026-04-02
> >
> > We thank the authors for the clarifications. Overall, the response clearly addresses the questions raised in my initial review. I recommend that the authors include the points discussed here (mainly explanation of the model cost and how the method could be adapted for even lower-resource languages) in the appendix.

---

> > > ### Author Response · Authors · 2026-04-07
> > >
> > > We sincerely thank Reviewer jsjx for the positive feedback and for confirming that our concerns have been fully resolved. We greatly appreciate your suggestion to enhance the transparency of our work. As recommended, we will add a new section in the Appendix detailing:
> > >
> > > - A comprehensive breakdown of the model training and inference costs.
> > > - An extended discussion on how our method can be adapted for even lower-resource settings.
> > >
> > > These additions will ensure that readers can better assess the efficiency and adaptability of our approach.  Thank you again for your constructive guidance, which has helped strengthen the completeness of our paper.

---

### Official Review · Reviewer_WLdp · 2026-03-13

**Soundness:** 2
**Presentation:** 2
**Significance:** 2
**Originality:** 2
**Overall Recommendation:** 4
**Confidence:** 4

**Summary:**

This paper proposes LiRA, a framework that enhances large language models for low-resource languages by anchoring multilingual inputs to a shared English semantic space. It integrates two components: Arca for representation alignment and LaSR for cross-lingual task consistency. Supported by theoretical bounds and evaluated on a new multi-language e-commerce dataset (LazRetrieval), LiRA achieves state-of-the-art results across diverse retrieval and reasoning tasks.

**Compliance With Llm Reviewing Policy:**

Affirmed.

**Final Justification:**

The detailed rebuttal has addressed my concerns, therefore  I have updated the score accordingly.

**Key Questions For Authors:**

1. How exactly does the proposed method operate on Chain-of-Thought (CoT) reasoning tasks? The overview figure lacks sufficient detail on this aspect, making it confusing to understand how the method processes and generates intermediate reasoning steps.
2. How sensitive is the framework to the choice of the LLM Critic, and would the performance hold if a non-Qwen model were used as the critic for a Qwen backbone?
3. Can you provide a clearer justification for evaluating the ablation components across disparate tasks rather than using a single, comprehensive benchmark?
4. How does the system perform in zero-shot cross-lingual transfer scenarios where no parallel data or translation candidates are available for the target language?
5. What is the practical impact of the meta-text leakage mentioned in the bad case analysis on the overall training stability, and what percentage of the data typically requires prompt engineering corrections?
6. Could you clarify how the empirical RKHS bound surrogates scale with models larger than 8B parameters, and whether the theoretical guarantees hold under extreme scaling?

**Limitations:**

Yes

**Strengths And Weaknesses:**

**Strengths:**
1. The theoretical foundation provides rigorous bounds on representation deviation and downstream stability under local Lipschitz continuity, which is a strong addition to empirical methods.
2. The introduction of the LazRetrieval dataset offers a valuable new benchmark for real-world, low-resource e-commerce retrieval across seven languages.
3. The proposed framework is modular and plug-and-play, demonstrating consistent improvements across multiple encoder architectures and varying parameter scales.


**Weaknesses:**
1. The core concept of aligning low-resource representations to an English semantic space lacks fundamental novelty, as this is a well-established paradigm in cross-lingual NLP.
2. The paper omits several recent and highly relevant works in cross-lingual reasoning, failing to discuss or compare against methods such as mCoT[1], MAPO[2], LinguaLIFT[3], and "Improving Multilingual Retrieval-Augmented Language Models through Dialectic Reasoning Argumentations[4]."
3. The theoretical analysis relies heavily on the assumption of an "ideal translation" reference point, which the authors admit is an unobserved mathematical construct, potentially weakening the practical applicability of the bounds.
4. The experimental setup predominantly uses the Qwen3 family for both the backbone and the critic model, introducing a potential architectural bias that might inflate the perceived effectiveness of the critic's evaluations.
5. The approach requires multiple translation models (up to four in the Pass@4 setting) during training and inference, which incurs significant computational overhead compared to standard fine-tuning pipelines.
6. While the paper claims to be effective for "low-resource" languages, the evaluations on LazRetrieval and public benchmarks mostly cover mid-resource languages (e.g., Vietnamese, Thai, Indonesian), lacking validation on truly extreme low-resource languages.
7. The ablation study evaluates different components on entirely different tasks (e.g., nDCG@10 on LazRetrieval, Pearson on STS22), which obscures how each component contributes to a single, unified objective.


[1] mCoT: Multilingual Instruction Tuning for Reasoning Consistency in Language Models

[2] MAPO: Advancing Multilingual Reasoning through Multilingual-Alignment-as-Preference Optimization

[3] LinguaLIFT: An Effective Two-stage Instruction Tuning Framework for Low-Resource Language Reasoning

[4] Improving Multilingual Retrieval-Augmented Language Models through Dialectic Reasoning Argumentations

---

> ### Author Rebuttal · Authors · 2026-03-30
>
> Thanks for your helpful review! Below are our clarifications
>
> **Novelty(W1)** We do not consider the introduction of the concept of “alignment of low-resource representations with the English semantic space” to be our core innovation. As reviewers jsjx and dE2m have pointed out, our core innovations are: the explicit mathematical modeling of anchoring error and translation distortion rather than treating cross-language adaptation as a black-box fine-tuning problem. By late 2025, no prior study offered our theoretical guarantees or mathematically explained why dual-path concatenation—aligning low-resource vectors with the English semantic space and combining them with translation vectors—is superior. This fundamentally distinguishes our work. Crucially, our architecture and training paradigm distinct from recent works like MindMerger (NeurIPS 25) and Lusifer (SIGIR 25), underscoring our novelty
>
> **Related works(W2)** We acknowledge missing these related works and will include comparisons with mCoT, MAPO, LinguaLIFT in the revision. On MGSM, LiRA (71.1) outperforms mCoT (65.9), MAPO (58.0) and LinguaLIFT (65.5). On X-CSQA, LiRA (65.5) surpasses LinguaLIFT (61.5). Crucially, while these methods focus primarily on reasoning, our work provides extensive validation across retrieval and ranking tasks
>
> **Ideal translation(W3)?** We believe that this is merely a theoretical analysis; it doesn’t mean that we must find this ideal translation in reality, just as physicists assume the existence of “particle”. Our theoretical objectives and LiRA are, in fact, closely intertwined: our goal is to minimize translation error and alignment error, not to identify the vector representing this ideal translation. Furthermore, our loss function is designed based on these two theoretical error models, and LiRA has demonstrated the validity of the theory
>
> **Qwen family?(W4, Q2)** Disentangling family bias from Qwen3's inherent superiority is infeasible, as Qwen3 demonstrably excels in translation quality and instruction-following. Replacing the Critic would not isolate bias, as performance gaps between critics would confound results. Given Qwen3's SOTA status in open-source, such experiments might be uninformative. Analogously, a talented student's success cannot be attributed solely to the relation with the teacher. Thus, replacing the Critic reveals only model performance gaps, not bias
>
> **Time cost(W5)** Appendix D.3 confirms translation accounts for most overhead; offline processing eliminates training/inference delays. On STS22 (single A100), Arca trains in minutes per epoch and LaSR in 10 minutes. Given LaSR requires <1 epoch, efficiency is high. Moreover, pre-processed translations represent a one-time cost that is reusable, saving resources for future research
>
> **Low-resource language(W6)** As stated in main contributions, our goal is to transfer English LLM capabilities to mid/low-resource languages. Accordingly, we evaluated across all resource levels (e.g.,  PK-Urdu in LazRetrieval and Bn, Sw in MGSM are considered low-resource languages in previous studies; Malay, Vietnamese, Filipino, Thai in LazRetrieval are considered mid-resource languages). Generally, most results show improvements following the order: low-resource > medium > high-resource, confirming LiRA is more effective for low-resource languages while enhancing performance across the board. For detailed discussion, please refer to our response to Reviewer jsjx, Point 3
>
> **Ablation(W7,Q3)** We use task-specific metrics for retrieval, ranking, and reasoning to ensure fairness. Table 5 shows consistent trends: removing components like the LLM Critic causes the largest drop across all tasks, confirming their crucial role in cross-lingual representation. We will elaborate in the main text
>
> **CoT?(Q1)** Our task definitions and experimental setups do not include any CoT-related components, and in our experiments on MGSM and X-CSQA, we set the parameter `enable_thinking=False`. We are more interested in the capabilities of the LLMs themselves rather than the CoT process
>
> **Q4** LiRA does not rely on readily available parallel translations in the target language, as we prepare them offline using translation models or LLMs, thus representing a zero-shot scenario. If you mean disallowing the use of any translation models to prepare parallel data, then we cannot achieve alignment between the target language and the English space, which contradicts LiRA setup
>
> **Q5** In our analysis, we noted that prompt engineering significantly mitigates meta-text leakage. Furthermore, Arca's Actor module, leveraging LLM and Embedding Critics for selection, effectively filters out nearly all such flawed translations. This further demonstrates that the Actor mechanism enhances training stability
>
> **Q6** The largest Qwen3-E model is 8B; with no larger variants, we can only measure the 8B upper bound. The theoretical guarantee always holds, as this RKHS bound exists for any finite dataset

---

> > ### Author Rebuttal · Reviewer_WLdp · 2026-04-04
> >
> > Thanks for your detailed rebuttal, my concerns are addressed, and I have updated the score accordingly;

---

> > > ### Author Response · Authors · 2026-04-07
> > >
> > > Thank you very much for your follow-up and for taking the time to carefully read our rebuttal. We sincerely appreciate your positive acknowledgment that our responses have adequately addressed your concerns, as well as your decision to update the score accordingly. Your constructive feedback on novelty, evaluation scope, theoretical assumptions, and practical efficiency has been extremely valuable in helping us improve the paper. We are truly grateful for your careful reconsideration and support.

---

### Official Review · Reviewer_3Zkd · 2026-03-14

**Soundness:** 2
**Presentation:** 3
**Significance:** 2
**Originality:** 2
**Overall Recommendation:** 3
**Confidence:** 3

**Summary:**

This paper aims to improve cross-lingual adaptation by introducing LiRA, a lightweight fine-tuning approach built on top of existing pretrained models that jointly optimizes representation stability and cross-lingual consistency. The method aligns low-resource language representations with English representations so that the model can retain reasoning capabilities learned in English.
The approach concatenates the English embedding of a sentence x with the multilingual embedding of its translation y in a low-resource language (i.e., z = [g(x); h(y)]). The objective is to minimize the distance between this vector and a reference vector z* = [h(y*); h(y*)], where y* is the correct translation of y. In other words, the model is trained to minimize |z − z*|.
In practice, the framework uses a language model as the translator and another LLM as a critique function, and applies reinforcement learning to fine-tune the model.
For evaluation, the authors introduce a new multilingual product retrieval dataset covering five Southeast Asian and two South Asian languages. They also evaluate on several existing benchmarks, including BelebeleRetrieval, MLQARetrieval, STS22 (text similarity), MGSM (math reasoning), and X-CSQA (reading comprehension). The reported results show that the fine-tuned model improves performance across most datasets and languages.

**Compliance With Llm Reviewing Policy:**

Affirmed.

**Final Justification:**

There are still claims made in the paper I am not convinced of, but missing details/details I had missed were clarified in the rebuttal.

**Key Questions For Authors:**

In what sense is the method lightweight, given that the total number of parameters increases substantially (e.g., from 8B to 15.9B)? That is, do the observed improvements stem from the proposed method itself or from the additional parameters and potentially large-scale pretraining in similar domains?
As the approach relies on machine translation, how can we be confident that it will be effective for genuinely low-resource languages where translation quality is poor? Has there been any analysis of the impact of MT quality on results?
The claim is made that “despite having more parameters, the inference time remains similar”: this is a surprising claim, so is there empirical evidence to support this?

**Limitations:**

yes

**Strengths And Weaknesses:**

== Overall Review:
Although the proposed method shows improvements for non-English languages, not all of the evaluated languages are truly low-resource. It is also somewhat misleading to describe the method as lightweight, given that the total number of parameters increases substantially (e.g., from 8B to 15.9B). It is therefore unclear whether the observed improvements stem from the proposed method itself or from the additional parameters and potentially large-scale pretraining in similar domains. Furthermore, since the approach relies on machine translation, its effectiveness for genuinely low-resource languages remains uncertain, particularly in settings where translation quality is poor.
While the idea is interesting, I am not fully convinced that the method is properly evaluated, especially given the lack of fair comparison in terms of model parameter size. The authors also claim that “despite having more parameters, the inference time remains similar,” which is difficult to accept without further experimental evidence. In addition, although the overall structure of the paper is reasonable, the presentation of technical details could be improved.
Overall, I do not think this paper is ready for publication at ICML.

== Strengths:
- The paper addresses an important problem in cross-lingual adaptation, particularly the challenge of transferring reasoning capabilities from English to other languages.
- The authors introduce a new multilingual product retrieval dataset covering several Southeast Asian and South Asian languages, which could be a useful resource for future research.
- The experimental evaluation includes a diverse set of benchmarks across different tasks (retrieval, text similarity, math reasoning, and reading comprehension), providing a relatively broad view of the model’s performance.

== Weaknesses

- The reported improvements appear to rely on comparisons between models with substantially different parameter sizes. For example, Qwen-3-8B is compared with LiRA-Base, which has around 14.4B parameters. Given this large increase in model size, it is difficult to view the method as a lightweight approach, especially when the performance gains are relatively modest.
- The technical presentation is difficult to follow, and several important details are missing. For instance, the choice of using an LLM judge as a translation critic is not well justified. It is also unclear why LiRA-Base and LiRA-Large use Qwen3-1.7B as the critic while LiRA-Max uses Qwen3-32B.
- The paper claims effectiveness for low-resource languages, but many of the evaluated languages appear to be medium- to high-resource (e.g., Japanese, Chinese, Portuguese). It would be better to clearly specify the language categories and align the claims accordingly.
- The reported improvements on the new retrieval dataset (LazRetrieval) are not fully convincing, since the model is trained with around 1,000k examples per language. The details of the training setup for this dataset and others are also not clearly described.

---

> ### Author Rebuttal · Authors · 2026-03-30
>
> Thanks for your helpful review! Below are clarifications.
>
> **Lightweight?** Our 'lightweight' claim refers strictly to trainable parameters. While translation models increase the total count (e.g., 4.3B in LiRA-Base vs. 1.7B Critic), they are excluded from training and inference overhead as translation is pre-computed offline. Moreover, open-sourcing these pre-computed translations ensures this one-time cost is highly reusable across experiments, saving computational resources for future research.
>
> **Technical presentation** Detailed technical backgrounds are provided in Appendices A.1–A.3 (Proofs, Assumptions, Representation) and Sections 3 & 3.3 (Motivation & Theory-Practice connection). We selected Qwen3 as the critic since it was the leading open-source LLM at the time (per Qwen's technical report). Crucially, the critic's parameter count must not be less than the translator's to prevent evaluation decline caused by inferior critic performance. Consequently, when scaling the translator (e.g., via LiRA_MAX), we correspondingly increase the discriminator's parameters.
>
> **Language categories** As stated in main contributions, our goal is to transfer English LLM capabilities to mid/low-resource languages. Accordingly, we evaluated across all resource levels (e.g.,  PK-Urdu [1], BD-Bengali in LazRetrieval and Bn-Bengali, Sw-Swahili in MGSM [2] are considered low-resource languages in previous studies; Malay, Vietnamese, Filipino, Thai in LazRetrieval are considered mid-resource languages). High-resource languages were also evaluated to ensure no performance degradation. Generally, most results show improvements following the order: low-resource > medium > high-resource, confirming our method is more effective for low-resource languages while enhancing performance across the board.
>
>  **Dataset** As stated in Section 5.2, Dataset, our experiments use LazRetrieval (10k samples) unless otherwise noted (line 373 right column, page 7). We have clarified that our method uses LazRetrieval, not LazRetrieval-mega (100k samples). We open-sourced LazRetrieval-mega to facilitate the preparation of pre-training data for other researchers, not for training our model. Detailed training configurations and experimental settings are provided in Appendix Sections C and D. Please refer to the appendix. Thank you.
>
> **Gain from additional parameters?** Simply combining pre-trained models (increasing model parameters) does not improve performance; on the contrary, performance falls below the baseline because the dual-path pre-trained representations are not aligned without training. Improvements are only achieved after applying our method. We will include these experiments in the appendix.
>
> **Genuinely Low-resource Languages** First, we have validated our model's effectiveness on languages considered low-resource in previously studies (e.g., Urdu, Bengali). Second, our approach does rely on the pretrained model's ability to recognize and encode the target language. We assume that by "extremely low-resource" languages you are referring to those for which even usable data are scarce (e.g., African tribal languages). If pre-trained models cannot recognize or encode these languages, or if no usable corpora exist, our method is unlikely to perform well without adaptation. To address this, we can replace the pretrained backbone with a model capable of encoding these languages—for instance, by first fine-tuning the pretrained model on available collected corpora to establish basic recognition. Our method can then transfer capabilities learned from high-resource languages to these low-resource targets. This adaptability constitutes the core value of our approach.
>
> **Impact of MT quality** Our paper already provides relevant studies regarding the impact of translation quality. Appendix D.3 compares models using small-parameter translation models with low-quality translation against those leveraging larger LLMs, while Section 5.3 demonstrates that scaling the translation model (and thus improving quality) enhances our method's efficacy. We posit this result can be reasonably extrapolated to languages with poorer translation that you mentioned. Since quantifying translation quality is not the focus of our work, we merely conduct a manual evaluation of translation showcases and verify that translation quality indeed improves as the parameter count of the translation model increases, which aligns with general empirical expectations.
>
> **Inference Time** Appendix D.3 confirms translation accounts for most overhead; offline processing eliminates training/inference delays. On STS22 (single A100), Arca trains in minutes per epoch and LaSR in 10 minutes. Given LaSR requires <1 epoch, efficiency is high. Moreover, pre-processed translations represent a one-time cost that is reusable, saving resources for future research.
>
> [1] Building and Evaluating a High Quality Parallel Corpus for English Urdu Low Resource Machine Translation.
>
> [2] MindMerger

---

> > ### Author Rebuttal · Reviewer_3Zkd · 2026-04-03
> >
> > Thanks for the detailed response. I have upgraded my rating slightly based on this additional information. A couple of additional responses:
> >
> > - re low-resource, there is a big gap between languages like Urdu and Bengali and African tribal languages. One reference that the authors may find instructive in terms of the (mis)use of the term low-resource is:
> >
> > https://aclanthology.org/2024.emnlp-main.983.pdf
> >
> > but there are plenty of languages which are meaningfully low-resource in terms of there being perhaps millions of tokens rather than 100s of millions or billions of tokens of corpus text that would have been more compelling.

---

> > > ### Author Response · Authors · 2026-04-07
> > >
> > > We sincerely thank the reviewer for this insightful comment and for bringing the instructive analysis and discussion by Nigatu et al. (2024) to our attention. This paper offers highly profound insights, and we will cite it in the Limitations section and provide further discussion.
> > >
> > > **1.** Indeed, as the paper points out, the definition of “low-resource” should be examined along far more dimensions than data volume alone. Although languages such as Urdu, Bengali, and African languages like Swahili are often treated in many studies as representative “low-resource languages,” there remain many endangered African tribal languages whose challenges go far beyond simple data scarcity, lacking even a basic digital presence. These languages are systematically marginalized across multiple dimensions, including socio-political status, human and digital resources, language technology artifacts, and community agency. Addressing these issues requires the collective effort of researchers across the field, with a genuine commitment to community needs. This includes building stronger data infrastructure, fostering local linguists and research ecosystems, and establishing community-led technological governance and ethical norms, so as to create a more equitable and inclusive language technology community. This is also precisely the original motivation behind our research.
> > >
> > > **2.** As noted above, although Urdu and Bengali do possess some amount of raw corpora, in the domain of product retrieval, these South Asian languages still face a severe shortage of high-quality, structured, annotated data, which directly constrains the performance of related retrieval and recommendation models. At the same time, however, e-commerce penetration in these regions is rapidly increasing and has become an indispensable part of everyday life for local residents. This striking mismatch between growing practical demand and lagging technical support is exactly why we decided to open-source this product dataset. By providing benchmarks for training and evaluation, we hope to fill this critical gap and thereby promote deeper research and practical deployment in this area.
> > >
> > > **3.** It is also worth noting that, on e-commerce platforms, we observed a structural challenge faced by Urdu that is highly similar to the situation described by Nigatu et al. (2024) for endangered languages such as Numma-guhooni and Konkani. Owing to historical factors and the inertia of digitization, product titles and user queries on these platforms exhibit a strong English-dominant pattern, which has led to the systematic marginalization of Urdu as a local language in digital commercial spaces. To address this issue, we adopted targeted cleaning strategies during dataset construction, proactively filtering out purely English samples and preserving as much genuinely Urdu data as possible, so as to ensure that our study can faithfully reflect and tackle the challenge of modeling Urdu independently, without reliance on English.
> > >
> > > We truly appreciate your willingness to re-evaluate our work and your positive acknowledgment of our responses. Your insights have been invaluable in refining our paper, and we look forward to your continued support as the final decision process proceeds.

---

### Decision · Program_Chairs · 2026-04-30

**Decision:**

Accept (regular)

**Comment:**

This paper proposes LiRA, a framework that enhances large language models for low-resource languages by anchoring multilingual inputs to a shared English semantic space. It integrates two components: Arca for representation alignment and LaSR for cross-lingual task consistency. Supported by theoretical bounds and evaluated on a new multi-language e-commerce dataset (LazRetrieval), LiRA achieves state-of-the-art results across diverse retrieval and reasoning tasks.

Reviewers are generally positive about the paper especially due to the impressive performance on retrieval tasks and good performance over strong baselines like MindMerger. However, there are some concerns:
1) Unlike what is stated in the paper, the framework does not seem lightweight (Reviewer 3Zkd), it almost double the number of parameters of Qwen 8B to Qwen+Lira, at 15.6B. I believe a more fair comparison is to add Qwen 3 14B in the evaluation, which was skipped.

2) The approach is rather cumbersome, despite the reported improvement.

3) Evaluation does not cover low-resource languages, this is very important issue. Given the framework is mostly designed for low-resource languages, this is a limitation. Authors should expand evaluation to low-resource languages benchmark like AfriMGSM (for African languages), since mid-resource languages like Swahili are well supported by LLMs, so, the benefit of this model stacking approaches is limited.

4) Does the MindMerger and LIRA use the same encoder and decoder? Otherwise, evaluation is misleading.